# Cold-shock proteome of myoblasts reveals role of RBM3 in promotion of mitochondrial metabolism and myoblast differentiation
Paulami Dey[1,2], Srujanika Rajalaxmi[1], Pushpita Saha [1], Purvi Singh Thakur[1], Maroof Athar Hashmi[1,3], Heera Lal[1,3], Nistha Saini[1], Nirpendra Singh[1] & Arvind Ramanathan [1] ✉

Adaptation to hypothermia is important for skeletal muscle cells under physiological stress and is used for therapeutic hypothermia (mild hypothermia at 32 °C). We show that hypothermic preconditioning at 32 °C for 72 hours improves the differentiation of skeletal muscle myoblasts using both C2C12 and primary myoblasts isolated from 3 month and 18-month-old mice. We analyzed the cold-shock proteome of myoblasts exposed to hypothermia (32 °C for 6 and 48 h) and identified significant changes in pathways related to RNA processing and central carbon, fatty acid, and redox metabolism. The analysis revealed that levels of the cold-shock protein RBM3, an RNA-binding protein, increases with both acute and chronic exposure to hypothermic stress, and is necessary for the enhanced differentiation and maintenance of mitochondrial metabolism. We also show that overexpression of RBM3 at 37 °C is sufficient to promote mitochondrial metabolism, cellular proliferation, and differentiation of C2C12 and primary myoblasts. Proteomic analysis of C2C12 myoblasts overexpressing RBM3 show significant enrichment of pathways involved in fatty acid metabolism, RNA metabolism and the electron transport chain. Overall, we show that the cold-shock protein RBM3 is a critical factor that can be used for controlling the metabolic network of myoblasts.

Hypothermic adaptation is a hallmark of hibernating animals, and mild hypothermia has been used for preserving tissue homeostasis in patients, including the brain[1,2] and heart[2–6]. During hypothermia, mitochondrial signaling maintains cellular energy homeostasis and promotes cellular survival[7]. It has been demonstrated that exposing human myoblasts to mild hypothermic conditions can improve their transplantation efficiency and survival[8]. This observation can be leveraged for applications of myoblasts in cell-replacement therapies and underlines the importance of uncovering comprehensive mechanisms that mediate hypothermic adaption.

In hibernating animals, the core temperature can be as low as 0–7 °C[9,10], and under therapeutic hypothermia, the temperatures range around 32 °C[11–13]. RBM3 (RNA-binding motif protein 3) is a cold-responsive protein[14–16] shown to play a key role during hypothermia. RBM3 is a member of the family of RNA binding proteins (RBP) which are

characterized by the presence of an RNA binding domain and an intrinsically disordered domain[17,18]. RBM3 is a highly conserved glycine-rich protein with a molecular mass of 17 kDa[19]. RBPs are known to control gene expression through the regulation of protein synthesis and post-transcriptional processes like RNA splicing[20,21], RNA stability[20,22] and RNA localization[23]. They are also known to affect cellular physiology in the context of numerous diseases, e.g., neurodegenerative[24,25], cardiovascular disorders[26], cancer[27,28], and hyperglycemia[26,29].

The temperature sensing by RBM3 is mediated by non-sense mediated decay (NMD) linked to alternative splicing of RBM3 mRNA[30]. At lower temperatures, a poison exon (E3a) positioned at exon 3 of RBM3 mRNA gets spliced, thus there is no degradation of RBM3 mRNA by NMD[30]. Another study in human iPSC-derived neurons suggests that an alternative splicing factor, heterogeneous nuclear ribonucleoprotein H1 (HNRNPH1)

[1]Institute for Stem Cell Science and Regenerative Medicine (inStem), GKVK—Post, Bellary Rd, Bengaluru 560065 Karnataka, India. [2]SASTRA Deemed University, Tirumalaisamudram, Thanjavur 613401 Tamil Nadu, India. [3]Manipal Academy of Higher Education, Manipal 576104 Karnataka, India. ✉e-mail: arvind@instem.res.in

enhances the splicing of the poison exon by strongly binding to G-rich motifs in RBM3 mRNA[31]. A few downstream effector proteins of RBM3 have been identified, for example, RTN3 in HEK293 and a mice model of prion disease[32]. RTN3 mediates the neuroprotective effects of RBM3 under hypothermia by promoting synaptogenesis, preventing neuronal loss, and rescuing neurodegenerative disease-dependent cognitive decline[32]. IGF2 mRNA binding protein 2 is another downstream protein of RBM3 which upon interaction with RBM3, increases Insulin Growth Factor 2 (IGF2) release, and promotes neuronal stem cell proliferation[33]. Another study in ischemia–reperfusion (I/R) condition of cardiomyocytes suggests that RBM3 binds to the raptor and regulates the mTOR pathway[34].

RBM3 has been implicated in neuroprotection and prevention of apoptosis[35–37]. The cytoprotective effects of RBM3 have been examined in the context of degenerative brain disorders. A study utilizing murine models of Alzheimer's and prion diseases demonstrates that the overexpression of RBM3 promotes synapse protection and regeneration[36,38–40]. In the context of skeletal muscles, RBM3 can inhibit atrophy in C2C12 myotubes[41]. Overexpression of RBM3 has been shown to promote survival against peroxide stress in myoblasts[40]. A recent study using C2C12 cells has shown that RBM3 may bind to the 3' UTR and coding regions of mRNA involved in various cellular processes like translational initiation and proteasomal degradation[42].

A comprehensive knowledge of proteomic changes during hypothermia, in skeletal muscle cells, is lacking. How hypothermia in skeletal myoblasts affects metabolic pathways and if RBM3 has a direct role in controlling hypothermia-associated metabolic rewiring remains to be further explored. To fill this gap in understanding, in this study, we performed a systematic analysis of hypothermic stress using untargeted proteomics. Our study reveals that in response to both acute and chronic hypothermia, myoblasts mount a response involving mitochondrial metabolism, RNA processing, and increasing levels of RBM3. We show that overexpression of RBM3 in myoblasts even at 37 °C is sufficient to promote mitochondrial metabolism, cellular proliferation, and differentiation in skeletal muscle myoblasts.

## Results

### Hypothermic adaptation enhances differentiation of C2C12 myoblasts

To test the effects of hypothermic adaptation on skeletal muscle differentiation, we grew C2C12 myoblasts at 32 °C for 72 h (37 °C was used as the control). This was followed by differentiation into myotubes (using 2% horse serum) at 37 °C for 7 days (Fig. S1a). We performed a western blot analysis where we observed that the protein levels of myosin heavy chain (MHC, a late differentiation marker[43,44]) in day 6 and day 7 myotubes, were significantly higher after hypothermic pre-conditioning as compared to the day 6 and day 7 of 37 °C control respectively. (Fig. 1a, b). We performed immunofluorescence analysis of C2C12 myotubes at day 6 by staining them with anti-MHC antibody and quantified the levels of differentiation. We observed that the fusion index and myotube diameter of cells pre-conditioned at 32 °C was significantly higher than 37 °C (Fig. 1c, d). This suggested that hypothermic adaptation enhances the differentiation of C2C12 myoblasts.

### Hypothermic adaptation improves differentiation of primary skeletal muscle myoblasts isolated from both 3-month-old and 18-month-old mice

Mouse primary skeletal muscle myoblasts were isolated from wild-type B6/J mice (3-month-old and 18-month-old mice respectively). Myoblasts were grown at 32 °C for 72 h and then differentiated at 37 °C for 4 days (37 °C was used as a control) (Fig. S1b). We performed immunofluorescence analysis of the myotubes (from 3-month-old mice) by staining the myotubes with anti-MHC antibody and quantified the levels of differentiation. We observed that the myotube diameter of cells exposed to hypothermic pre-conditioning was significantly higher compared to the 37 °C control whereas the myotube fusion index was lower than the control (Fig. 1e, f). We also performed a western blot analysis of protein extracted from the myotubes (day 0, 2, 3, and 4 of differentiation) which were differentiated from primary myoblasts isolated from 3-month-old mice. We observed that the protein levels of

MHC in day 4-myotubes under hypothermic pre-conditioning was higher as compared to the day 4 of 37 °C control (Fig. 1g, h). We considered the MHC levels of day 4-myotubes for further analysis and quantification. We also observed that the protein levels of MHC in myotubes exposed to hypothermic pre-conditioning were significantly higher than the 37 °C control in 18-month-old mice (Fig. 1i, j). We also observed that myoblasts isolated from both 3 and 18-month-old mice showed enhanced differentiation upon hypothermic per-conditioning as judged by MHC levels quantified at day 4 (Fig. S1c). These results show that hypothermic adaptation in mouse skeletal myoblasts enhances differentiation in both 3-month-old and 18-month-old mice.

### C2C12 myoblasts subjected to acute (6 h) hypothermic stress (32 °C) cause proteomic changes in cellular processes associated with chromatin organization, cytoskeleton, RNA, and fatty acid metabolism

To comprehensively map pathways that respond to acute hypothermic stress (6 h) we performed proteomic analysis of C2C12 myoblasts cultured at 32 °C and 37 °C for 6 h, respectively. The dataset generated from proteomic analysis was processed based on two-step filtering criteria. The first step was based on fold change (fold change value ≥1.5 were considered to be upregulated and ≤0.6 was considered to be downregulated). In addition to fold change, we also calculated $p$ value for all the proteins. Our analysis showed that among the 3813 proteins detected, 1020 proteins were upregulated, 2676 proteins were downregulated, and 117 proteins were unchanged at 6 h compared to the 37 °C control (Fig. S2e). We performed a GO pathway enrichment analysis using Metascape[45] (Fig. 2a) which showed that multiple cellular processes were upregulated and downregulated significantly compared to the 37 °C control, upon acute hypothermic stress. Pathways associated with chromosome condensation were one of the most significantly upregulated pathways along with cytoskeletal reorganization. We also observed that pathways involved in matrix metalloproteases and ECM remodeling were significantly upregulated. Pathways associated with IL-17 signaling were also enriched in the GO analysis. We next performed a protein-protein interaction analysis of the proteomics data of C2C12 myoblasts at 32 °C for 6 h using Metascape (Fig. 2b) and observed significant interactions between histone component proteins (H2AC33, H2AC26, H2AC40, H2AC39, etc.), and their levels were significantly higher as compared to the 37 °C control myoblasts (Fig. S2a). We also observed interactive networks between intermediate filament organization, intermediate filament cytoskeleton organization, formation of the cornified envelope, and integrin cell surface interactions (Fig. 2b). Proteins involved in muscle function and development (ACTA1, ACTA2, FHL2, FHL3, etc.) also showed significant interactive network (Fig. 2b).

Using GO pathway enrichment analysis, we found proteins with decreased abundance which were associated with RNA metabolism, fatty acid and glutathione metabolism as compared to 37 °C control (Fig. 2c). A protein-protein interaction analysis revealed significant interactions between proteins involved in translation (RPL35, MRPL17, MRPL2, etc.), transport of mature mRNA (NUP153, NUP58, NUP155 etc.), RNA pol II transcription elongation (POLR2E, POLR2L, etc.), RNA splicing and spliceosome complex (DDX46, DDX17, SF3A2, etc.) respectively (Fig. 2d, S2b). Significant interactions were also observed among proteins involved in fatty acid elongation, biosynthesis of unsaturated fatty acid, and very long-chain fatty acid metabolic process (ACOT1, ACOT2, ACOT3, etc.) (Fig. 2d).

### C2C12 myoblasts subjected to chronic (48 h) hypothermic stress (32 °C) show proteomic adaptations in cellular processes associated with carbon, redox metabolism and RNA processing

To understand the pathways associated with response to chronic hypothermic stress, we performed proteomic profiling of C2C12 myoblasts cultured at 32 °C and 37 °C for 48 h, respectively. We performed proteomic profiling using a SWATH proteomics workflow. Our analysis showed that among the 3813 proteins detected, 996 proteins were upregulated at 48 h, 1320 were downregulated and 1497 were unchanged compared to 37 °C control

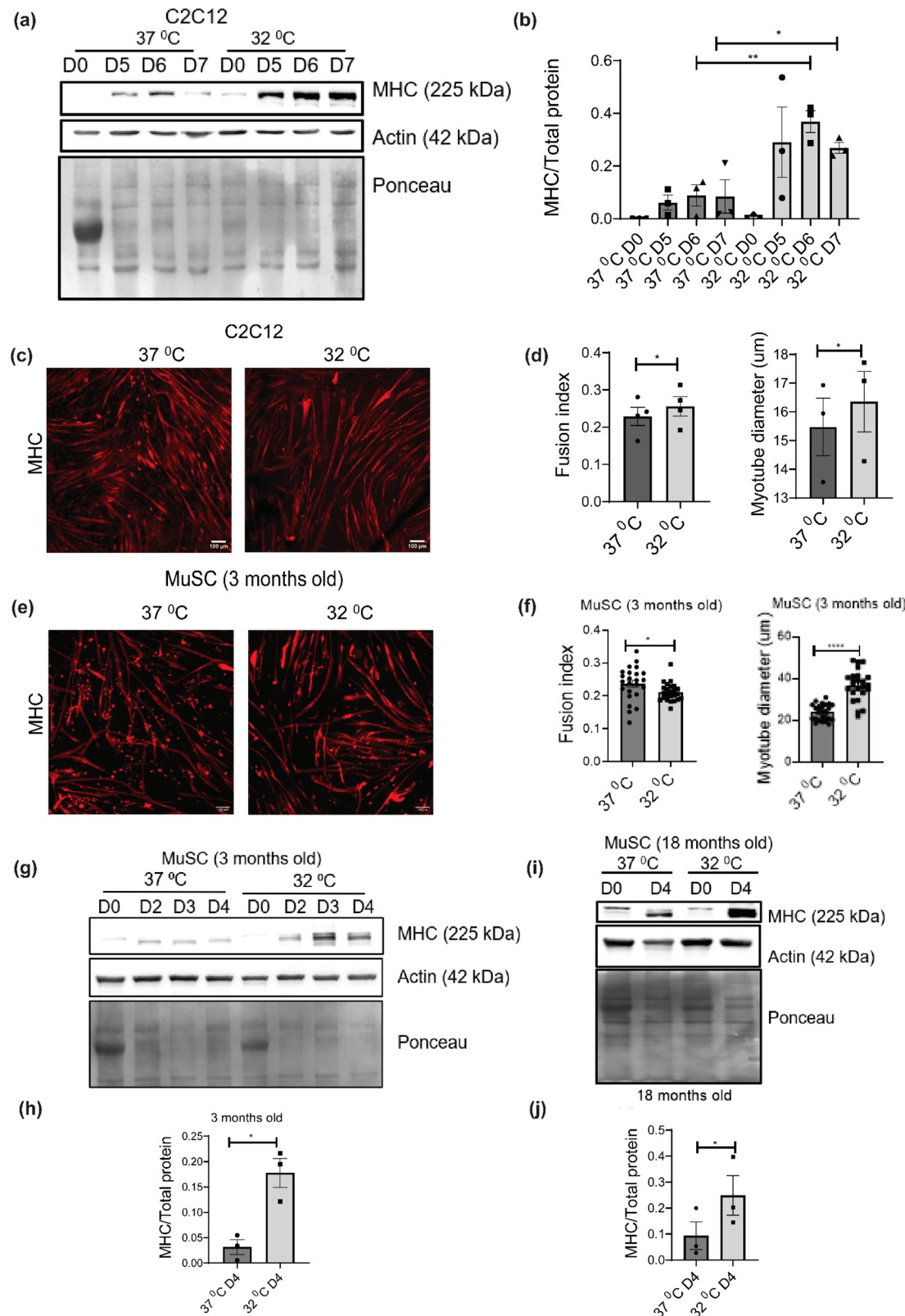

myoblasts (Fold change value ≥1.5 were considered to be upregulated and ≤0.6 were considered to be downregulated) (Fig. S2f). We performed a GO pathway enrichment analysis using Metascape[45] (Fig. 3a) which showed that multiple cellular processes were enriched significantly at 32 °C as compared to 37 °C control. The levels of proteins involved in Lipid metabolism, fatty acid biosynthesis, glutathione metabolic process, and monocarboxylic metabolic process were significantly higher at 32 °C compared to control.

Protein-protein interaction analysis using Metascape software revealed significant interactions among proteins involved in glutathione metabolism (GSTM1, GSTM5, GSTA4, etc.), peroxisomal import (ACOX3, PTGR3, DHRS4, etc.), ribosome biosynthesis and rRNA processing (RPP38, MRPS15, UTP14A, etc.) (Figs. 3b, S2c). The proteins involved in immunoregulatory interactions are most significantly upregulated by chronic hypothermia. In agreement with this, protein-protein interaction analysis

**Fig. 1 | Hypothermic pre-conditioning of skeletal muscle myoblasts at 32 °C for 72 h followed by differentiation. a** Western blot of protein levels of myosin heavy chain (MHC) in C2C12 cells differentiated for 5, 6, and 7 days at 37 °C after hypothermic pre-conditioning (**b**) Bar graph quantifying the western blot in a, where y-axis represents the intensity of MHC normalized to total protein ($n = 3$). Significance tested by two-tailed student's *t* test (unpaired). **c** C2C12 cells differentiated for 6 days at 37 °C after hypothermic preconditioning and immunostained with MHC antibody. **d** Bar graph showing the myotube fusion index ($n = 4$) and myotube diameter quantified from C2C12 myotubes shown in (**c**) ($n = 3$). Significance tested by two-tailed student's *t* test (paired). **e** Primary myoblasts (from 3-month-old mice) differentiated for 4 days at 37 °C after hypothermic pre-conditioning and immunostained with MHC antibody. **f** Bar graph showing the myotube fusion index and myotube diameter quantified from (**e**) (24 individual data points taken from 3 separate experiments). Significance tested by two-tailed student's *t* test (unpaired). **g** Western blot of protein levels of MHC using primary myoblasts (from 3-month-old mice) differentiated for 4 days at 37 °C after hypothermic pre-conditioning. **h** Bar graph quantifying the western blot in (**g**), where the y-axis represents the intensity of MHC normalized to total protein, ($n = 3$). Significance tested by two-tailed student's *t* test (unpaired). **i** Western blot of protein levels of primary myoblasts (from 18-month-old mice) differentiated for 4 days at 37 °C after hypothermic preconditioning. **j** Bar graph quantifying the western blot in i where the y-axis represents the intensity of MHC normalized to total protein ($n = 3$). Significance tested by two-tailed student's *t* test (paired). Error bar in the bar graphs represents standard error of mean (SEM). \*, \*\*, \*\*\*$p < 0.05$, 0.01 and 0.001.

reveals the importance of proteins involved in antigen processing and presentation (H2-K1, H2-Q7, H2-D1, H2-Q10, etc.) (Fig. 3b).

A number of proteins showed a decrease in abundance during chronic hypothermia. We performed a GO pathway enrichment analysis using Metascape which revealed that the levels of proteins involved in the pathways like intermediate filament organization, mRNA processing, and histone modification were significantly lower at 32 °C compared to the 37 °C control (Fig. 3c, S2d). Protein-protein interaction studies revealed significant interactions of downregulated proteins involved in various cellular processes like mRNA processing (HNRNPDL, DDX27, DDX39A, and RALY), CREB phosphorylation, and activation of CAMK (CALM1, CALM2, CALM3, VAPA, etc.), Ras signaling pathway (MAPK9, MAPK10, CDC16, etc.). Also, proteins involved in intermediate filament organization (Keratin proteins), histone modification (HDAC3, HDAC1, RNF20, etc.), mRNA splicing (NUP37, SNRPB, SNRPN, etc.) were significantly lower at 32 °C compared to the 37 °C control (Fig. 3d).

### Exposure of skeletal muscle myoblasts to mild hypothermic stress (32 °C) increases expression of the cold-shock protein RBM3

Proteomic analysis of mild hypothermia revealed two known cold shock proteins to be upregulated: RBM3[46] and CIRBP[46]. RBM3 was upregulated at both acute and chronic hypothermic stress (Fig. 4a), whereas CIRBP was not upregulated by acute hypothermic stress of 6 h, but it was upregulated at chronic hypothermic stress of 48 h as compared to control (Fig. S2i). To validate the increase in levels of RBM3 under hypothermia, we grew C2C12 myoblasts at 32 °C for 48 and 72 h, respectively (37 °C was used as control). We performed western blot analysis where we probed the blot with an anti-RBM3 antibody. We observed that the protein levels of RBM3 were significantly higher under hypothermic conditions compared to the control (Fig. 4b, c). We performed similar experiments using mouse primary skeletal myoblasts (3 and 18-month-old) and found that the protein levels of RBM3 were significantly higher under hypothermic conditions at 72 h as compared to 37 °C control in both 3 and 18-month-old mice (Fig. S3a, b).

### Hypothermic preconditioning (32 °C) for 72 h enhances differentiation of skeletal muscle cells in an RBM3-dependent manner

Since hypothermic preconditioning increased differentiation of skeletal muscle myoblasts and RBM3 was upregulated under hypothermia, we hypothesized that the two observations could be connected. To test this, C2C12 myoblasts were transfected with siRBM3 (scrambled siRNA was used as control) (Fig. S3c). After 30 h of siRNA treatment, we preconditioned the C2C12 myoblasts at 32 °C for 72 h. This was followed by differentiation of the myoblasts into myotubes at 37 °C for 7 days. We performed western blot analysis where we probed the blot with anti-MHC antibody. We used MHC levels at day 0 and day 6 for further analysis and quantification. The protein levels of MHC at day 6 were significantly higher at 32 °C in scrambled siRNA-treated myotubes as compared to the 37 °C scrambled siRNA-treated myotubes. Unlike the controls, we observed that myotubes treated with siRBM3 showed a lower protein level of MHC as compared to 32 °C scrambled siRNA-treated cells (Fig. 4d, e). This shows that hypothermic preconditioning requires RBM3 for its effect on myoblast differentiation.

### Maintaining levels of mitochondrial respiration of C2C12 myoblasts pre-conditioned at 32 °C is dependent on RBM3

Since the proteomic analysis revealed that hypothermia for 48 h increased the levels of proteins involved in mitochondria-related metabolic processes (lipid metabolism, oxidative stress, redox pathways, monocarboxylic acid, and nicotinamide metabolism), we investigated the connection between RBM3 and mitochondrial respiration after hypothermic pre-conditioning. We performed seahorse respirometric analysis of C2C12 myoblasts transfected with siRBM3 (scrambled siRNA was used as control). After siRNA treatment, we pre-conditioned the C2C12 myoblasts (siRBM3 transfected and scrambled siRNA transfected) at 32 °C for 72 h. Next, we measured the oxygen consumption rate (OCR) of the myoblasts, under basal conditions as well as in the presence of mitochondrial inhibitors (Fig. 4f). Compared with myoblasts grown at 37 °C, hypothermic pre-conditioning by itself did not alter the mitochondrial respiration (basal, maximum respiration, and spare respiratory capacity). But siRBM3 treatment during hypothermic pre-conditioning, significantly inhibited mitochondrial respiration (Fig. 4f). This shows that RBM3 is required for metabolic adaption during hypothermic pre-conditioning.

### Overexpression of the RBM3 at 37 °C promotes mitochondrial metabolism in skeletal muscle myoblasts as judged by Seahorse and MTT assays

We investigated whether RBM3 by itself was sufficient to affect mitochondrial metabolism. Therefore, we performed a Seahorse respirometric analysis of stable C2C12 myoblasts overexpressing RBM3 (cells over-expressing GFP were used as control) (Fig. 5a). We generated C2C12 stable line overexpressing RBM3 by transducing C2C12 cells with retrovirus containing pMIG-RBM3 and pMIG-GFP respectively. We observed that the basal respiration of C2C12 myoblasts overexpressing RBM3 was higher as compared to the control. The maximum respiration, spare respiratory capacity, and ATP-linked respiration were also higher in myoblasts overexpressing RBM3 as compared to that of the control (Fig. 5b). This suggests that RBM3 mediates an improvement in the oxygen consumption rate even at 37 °C. We next performed an MTT assay to measure the metabolic viability of myoblasts overexpressing RBM3 in the presence of mitochondrial and glycolytic inhibitors respectively. We observed that the sensitivity of myoblasts overexpressing RBM3 to oligomycin (mitochondrial complex V inhibitor) and 2DG (2-deoxyglucose, a competitive inhibitor of glucose) was lower as compared to the control. The $IC_{50}$ of oligomycin in myoblasts overexpressing RBM3 was 1167 nM whereas that in control myoblasts was 247 nM (Fig. S3d). The $IC_{50}$ of 2DG in myoblasts overexpressing RBM3 was 1.42 mM whereas that in control myoblasts was 0.77 mM (Fig. S3e).

### Overexpression of RBM3 at 37 °C increases the levels of citrate, succinate and acetyl-CoA in skeletal muscle myoblasts as judged by targeted metabolic profiling

Since we observed an increase in mitochondrial oxygen consumption rate in myoblasts overexpressing RBM3, we measured the intracellular levels of glycolytic and TCA intermediates respectively. We observed that the intracellular levels of pyruvate, lactate, malate, succinate,

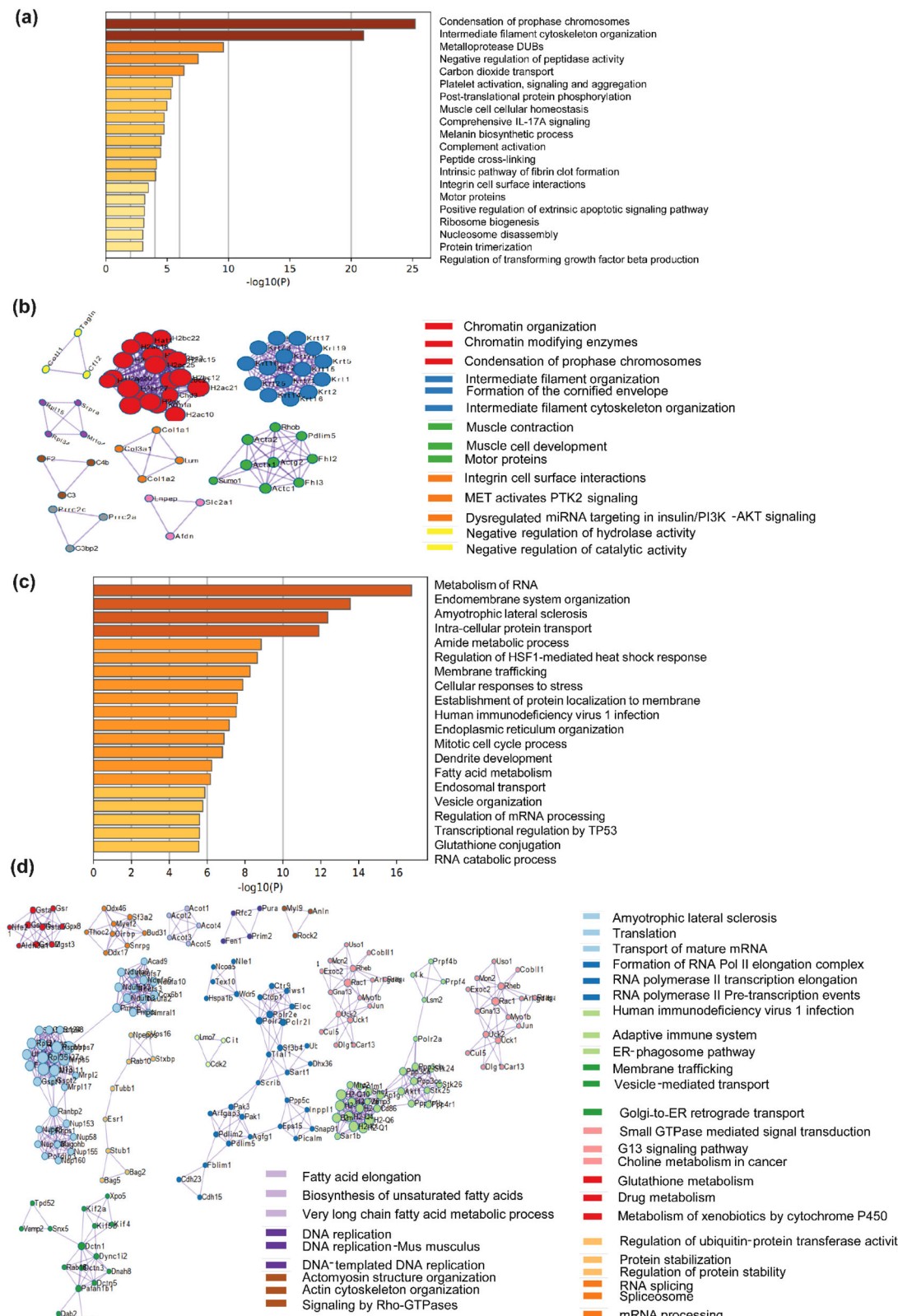

**Fig. 2 | Proteomic analysis of C2C12 myoblasts after 6 h of hypothermia at 32 °C.**
**a** GO enrichment pathway analysis of upregulated proteins at 32 °C. **b** GO
enrichment analysis of protein-protein interaction networks of upregulated proteins
at 32 °C (*n* = 3). **c** GO enrichment pathway analysis of downregulated proteins at
32 °C. **d** GO enrichment analysis of protein-protein interaction networks of
downregulated proteins at 32 °C (*n* = 3).

glutamate, and fumarate in cells overexpressing RBM3 were higher as
compared to the GFP control, among which citrate and succinate were
significantly higher (Fig. 5c, Fig. S4b, c). Intracellular ratio of lactate to
pyruvate was lower in myoblasts overexpressing RBM3 compared to

that in the control (Fig. S4a). The intracellular acetyl-CoA levels were
significantly higher in myoblasts overexpressing RBM3 compared to
that of the control (Fig. 5d). Concomitantly we observed that over-
expression of RBM3 was associated with higher protein levels of

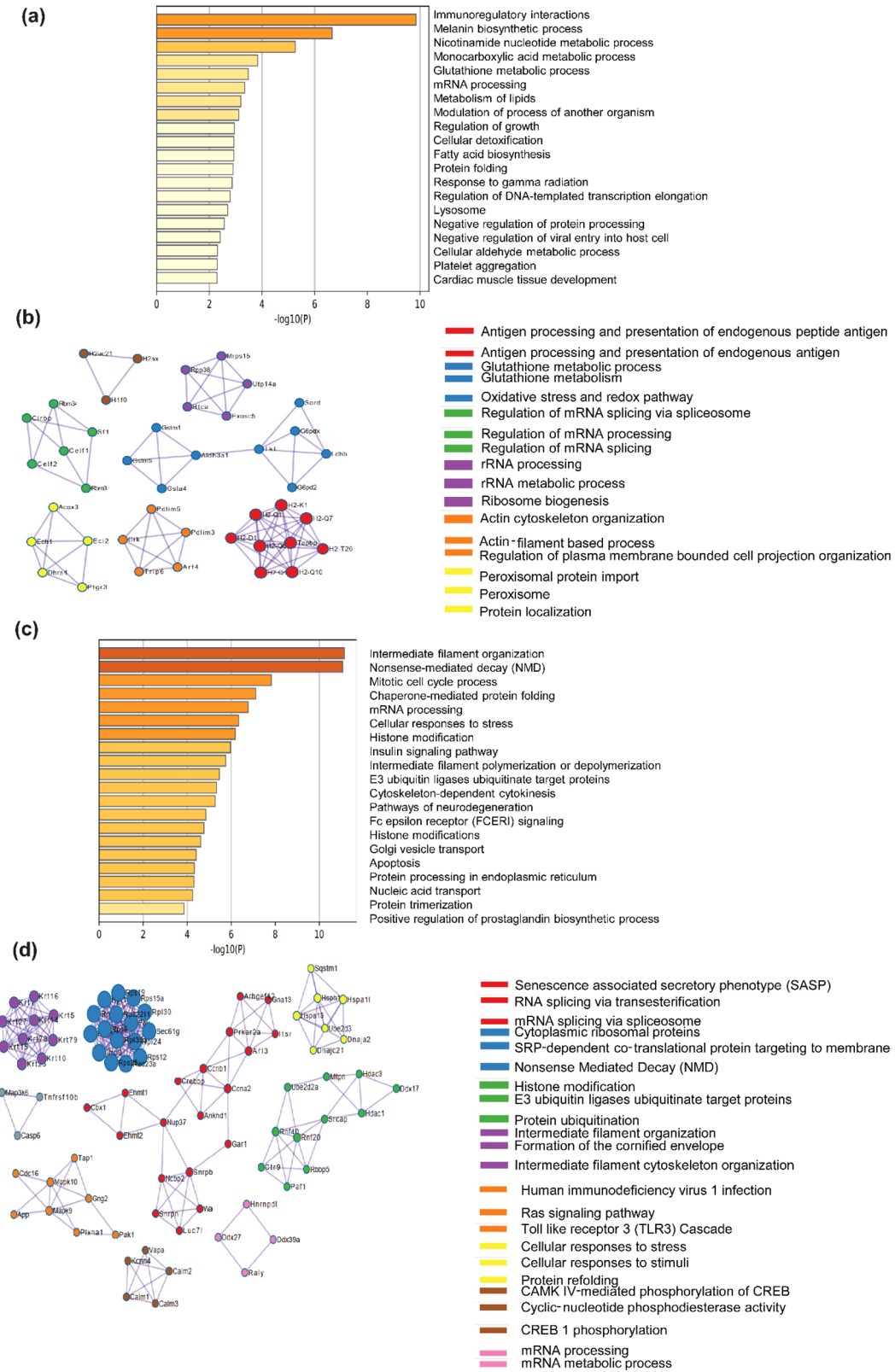

**Fig. 3 | Proteomic analysis of C2C12 myoblasts after 48 h of hypothermia at 32 °C. a** GO enrichment pathway analysis of upregulated proteins at 32 °C. **b** GO enrichment analysis of protein-protein interaction networks of upregulated proteins at 32 °C ($n$ = 3). **c** GO enrichment pathway analysis of downregulated proteins at 32 °C. **d** GO enrichment analysis of protein-protein interaction networks of downregulated at 32 °C ($n$ = 3).

pyruvate dehydrogenase (PDH) (Fig. S4d) and pyruvate kinase isoform 2 (PKM2) (Fig. S4e) whereas lower protein levels of pyruvate kinase isoform1 (PKM1) (Fig. S4e). These results show that RBM3 supports mitochondrial metabolism.

## Overexpression of RBM3 at 37 °C promotes proliferation of C2C12 myoblasts
Next, we tested the effects of RBM3 on cell proliferation and cell viability of mouse skeletal myoblasts. We grew the stable lines of C2C12 myoblasts

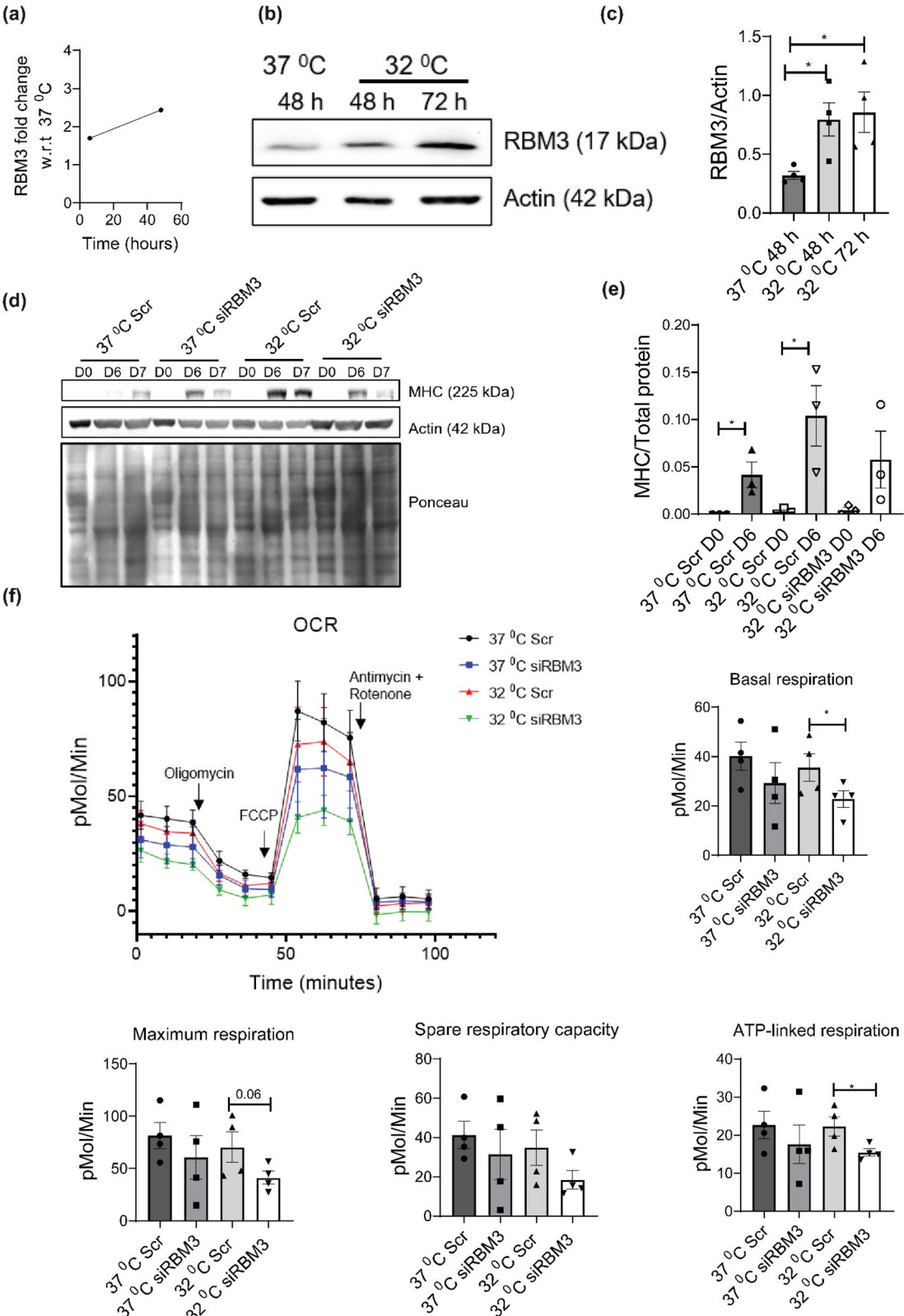

overexpressing RBM3-GFP and GFP (Fig. S5a) on coverslips respectively, stained them with EdU and performed epifluorescence microscopy to determine the % of EdU-positive nuclei. We observed that overexpression of RBM3 leads to a higher percentage of EdU-positive nuclei compared to the GFP control (Fig. 5e, f). We grew C2C12 myoblasts overexpressing RBM3-GFP and GFP on coverslips, stained them with Ki-67 antibody, and

performed confocal microscopy to determine the mean integrated intensity of the Ki-67 positive nuclei. We observed that the Ki-67 intensity was significantly higher in myoblasts overexpressing RBM3 as compared to the control (Fig. S5b, c). We also performed a cell counting assay where we plated an equal number of myoblasts (of both GFP and RBM3 respectively) and counted the cell number from day 1 to day 3. We observed that

**Fig. 4 | Hypothermic pre-conditioning of C2C12 myoblasts at 32 °C for 72 h followed by differentiation and respirometric analysis, with or without knockdown of RBM3. a** Line graph showing the fold change of protein levels of RBM3 upon hypothermic treatment at 32 °C with respect to 37 °C at different time points (6 h and 48 h). **b** Western blot of protein levels of RBM3 using C2C12 cells cultured at 37 °C and 32 °C for 48 h and 72 h respectively. **c** Bar graph quantifying the western blot image in b, where the y-axis represents the intensity of RBM3 normalized to Actin ($n = 4$). Significance tested by two-tailed student's $t$ test (unpaired). **d** Western blot of protein levels of MHC using C2C12 cells transfected with scrambled (Scr) or siRBM3 and differentiated for 6 days at 37 °C after hypothermic

pre-conditioning. **e** Bar graph quantifying the western blot image in d, where y-axis represents the intensity of MHC normalized to total protein, ($n = 3$). Significance tested by two-tailed student's $t$-test (unpaired). **f** Line graph showing the mitochondrial oxygen consumption rate (OCR) of C2C12 cells transfected with Scr or siRBM3 and incubated at 37 °C and 32 °C (72 h) respectively. X-axis represents time in minutes and the y-axis represents oxygen consumption rate in pMol/min. Bar graph quantifying the basal respiration, maximum respiration, spare respiratory capacity and ATP-linked respiration of the cells ($n = 4$). Error bar in the bar graphs represent standard error of mean (SEM). Significance tested by two-tailed student's $t$ test (unpaired). *, **, *** $p < 0.05$, 0.01 and 0.001.

overexpression of RBM3 leads to higher cellular proliferation in all three days compared to the control (Fig. 5h). Next, we tested whether the overexpression of RBM3 affected cell viability. We grew the C2C12 myoblasts in 96 well plates for two days and performed an MTT assay. We observed that C2C12 myoblasts overexpressing RBM3 has significantly higher cell viability as compared to the control cells overexpressing GFP (Fig. 5g). These observations suggest that overexpression of RBM3 promoted cell viability and proliferation of C2C12 myoblasts.

## Overexpression of RBM3 at 37 °C promotes differentiation of C2C12 and primary skeletal muscle myoblasts

Since RBM3 was required for hypothermia-driven enhanced differentiation of myoblasts, we tested whether RBM3 was sufficient for enhanced differentiation of myoblasts at 37 °C. Stable C2C12 myoblasts (overexpressing RBM3 and GFP respectively) were grown at 37 °C and differentiated at 37 °C for 7 days. We observed that protein levels of MHC was higher in C2C12 myotubes overexpressing RBM3 compared to the control. This indicates that RBM3 can promote the differentiation of myoblasts in a hypothermia-independent manner (Fig. 5i, j). We performed immunofluorescence with C2C12 myoblasts overexpressing RBM3 (myoblasts overexpressing GFP were used as control) followed by differentiation into myotubes at 37 °C for 6 days. We stained the myotubes with anti-MHC antibody and checked for the levels of differentiation. We observed that the fusion index and myotube diameter of cells overexpressing RBM3 were significantly higher as compared to the control (Fig. 5k, l).

We performed a similar experiment in primary mouse myoblasts where we electroporated these cells with pMIG-RBM3 (pMIG-GFP was used as control) plasmids and differentiated them at 37 °C for 4 days to form myotubes (Fig. S5d). We observed that the myotube diameter was significantly higher in cells overexpressing RBM3 compared to control. However, no significant change in the fusion index was observed (Fig. 5m, n). These results suggest that RBM3 is sufficient to promote differentiation in skeletal muscle cells.

## Proteomic analysis of stable C2C12 myoblasts overexpressing RBM3 shows rewiring of cellular metabolic processes

To understand the cellular processes controlled by RBM3, we performed a proteomic analysis of C2C12 myoblasts overexpressing RBM3 (GFP is used as control). Proteomic analysis showed that out of 1344 proteins detected, 302 were upregulated, 318 were downregulated and 724 were unchanged (Fig. 6a). There was a significant increase in protein levels involved in cellular processes like RNA metabolism, fatty acid metabolism, ETC in RBM3 overexpressing myoblasts compared to control, as analyzed by GO pathway enrichment analysis using Metascape (Fig. 6b). The levels of proteins involved in cellular processes like ribonucleoprotein-complex biogenesis, VEGFR2 signaling, and RNA localization were also significantly higher than control. A protein-protein interaction analysis performed using Metascape revealed significant interactions between proteins involved in cellular processes like PPAR signaling pathway (ACADL, ACADSB, ME2, CKB, etc.), ETC (NDUFV1, NDUFA5, PMPCB, etc.), transport of mature mRNA (POLDIP3, TPR, NUP160, etc.), VEGFR2 signaling (RAB11B, CDC42, CTNND1, etc.) and intermediate filament organization (Keratin proteins) (Fig. 6c, S6a, b, c). Also, protein-protein interaction analysis revealed significant interactions among proteins involved in RNA splicing

(SRRM2, SNRPB2, SF3B4, SNRPG, SNRP200, SF3B5), cleavage and polyadenylation (CPSF2), transcription initiation and regulation (DHX36, DHX9, HNRNPD), translation regulation (RPS27, RPS27A, RPS27L, RPL5, RPL8, RPS15), RNA degradation (PSMB7, PSMD6, CUL1, PSME2) (Fig. 6c).

A subset of proteins also showed decreased abundance after overexpression of RBM3. GO pathway enrichment analysis of this subset showed that proteins involved in RNA metabolism, tricarboxylic acid metabolism, Acyl-CoA metabolic process, and amino acid metabolism were significantly decreased in abundance (Fig. 7a). A Protein-protein interaction network analysis showed that the involvement of proteins in cellular processes like citric acid cycle (IDH3 and SDHA), ETC (ATP5H, NDUFB10, UQCRC2, COX4I1, etc.), mRNA processing (TRA2B, RNPS1, SRSF7, etc.), cytoskeleton organization (ACTR1A, ACTL6A, PDLIM7, etc.) and amino acid metabolism (PTPA, UAP1L1, etc.) (Fig. 7b). We next compared pathways that were commonly enriched between RBM3 overexpression and hypothermia, to understand cellular processes that might be involved in RBM3 mediated hypothermic adaptation. Venn diagram of the pathway enrichment analysis of proteins that were increased and decreased in levels driven by RBM3 overexpression and hypothermia are shown in Fig. S6e and, Fig. S6f, respectively. In the context of cellular metabolism, we observed that there was an enrichment of fatty acid metabolism and monocarboxylate processes among the pathways in S6e.

Using mouse neuroblastoma cells (N2a cell line), it was shown that RBM3 promotes global translation by associating with translating ribosomes[46]. Therefore, we investigated if overexpression of RBM3 increased the overall translation in C2C12 myoblasts. C2C12 myoblasts overexpressing RBM3 were treated with puromycin (1 µg/ml) for 30 min and 1 h respectively (C2C12 myoblasts overexpressing GFP were used as control) to measure changes in global protein synthesis. We performed a western blot analysis and probed the blot with an anti-puromycin antibody to quantify the levels of translating polypeptides. We did not observe any significant change in the levels of puromycin incorporation after both 30-min and 1-h treatment in C2C12 myoblasts overexpressing RBM3 as compared to the control (Fig. S7a, b, c). RBM3 has been suggested to promote global translation by increasing the phosphorylation of eIF4E (cap-binding protein required to initiate cap-dependent translation) in N2a cell lines[47]. Therefore, we investigated whether RBM3 affected the phosphorylation of eIF4E in C2C12 myoblasts. A western blot analysis was performed in C2C12 myoblasts overexpressing RBM3 (C2C12 myoblasts overexpressing GFP were used as control). We did not observe any significant change in the ratio of phosphorylated to the total level of eIF4E (Ser209) (Fig. S7d, e). However, we observed that the ratio of the phosphorylated to the total levels of 4E-BP1 (Tyr37/Tyr46) in C2C12 myoblasts overexpressing RBM3 was higher compared to that of the control (Fig. S7f, g). These results suggest that RBM3 does not cause a global increase in translation in C2C12 myoblasts but affects protein levels of only a subset of the proteome.

## Discussion

Cells adapt to hypothermic stress during numerous physiological situations, including hibernation and the underlying mechanisms are less understood. It is known that levels of numerous circulating metabolites, including lipids and ketone bodies are significantly increased during hibernation[48–50]. On the

**Article**

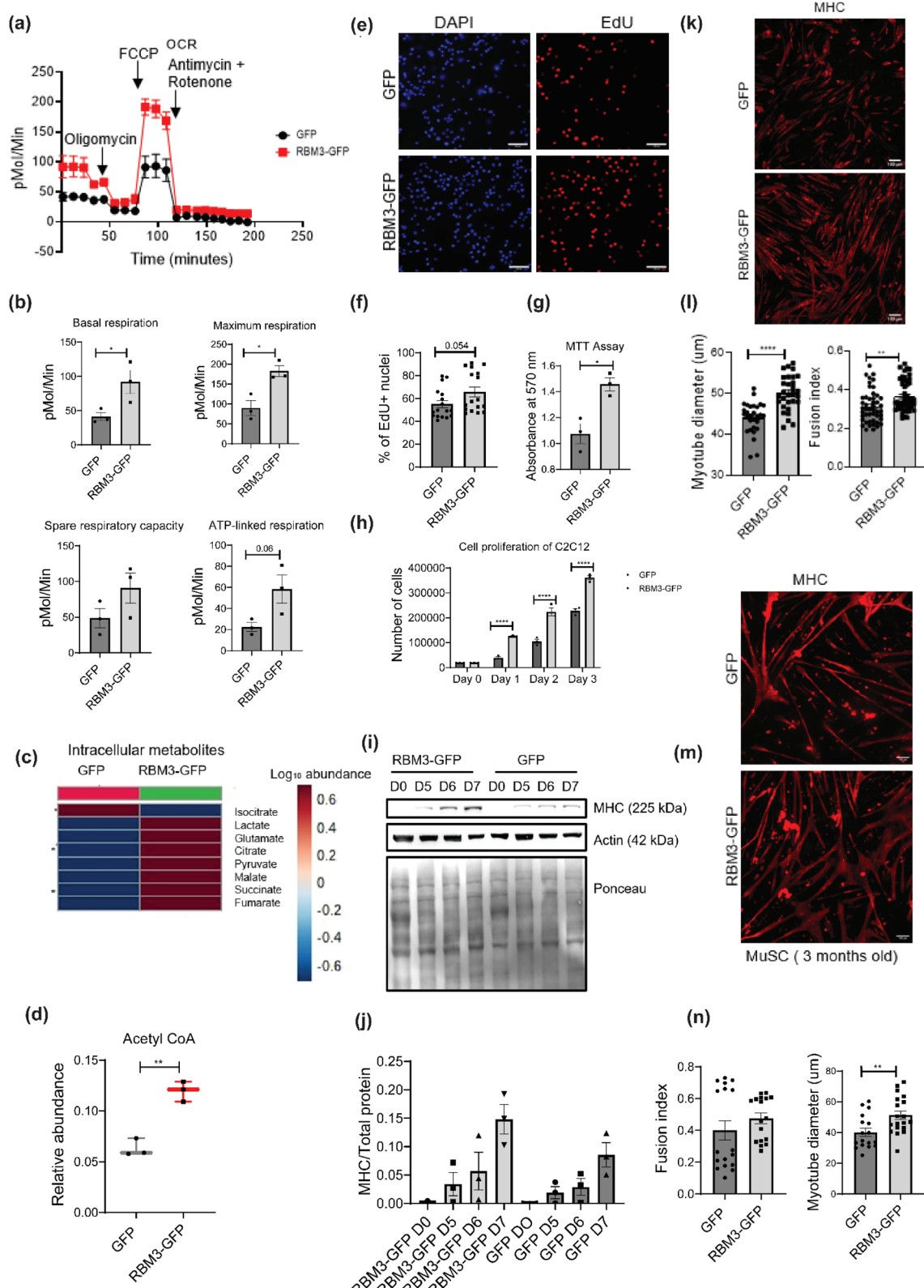

other hand, milder hypothermic stress (reducing the temperature of cells down to 32 °C) has been found to have therapeutic applications for preserving tissue homeostasis in cases of brain or cardiac injury[2–4]. Adaptation to hypothermia has been associated with improved skeletal muscle homeostasis[51]. Hypothermia is also known to improve the transplantation efficiencies of human satellite cells[8]. An important feature of hypothermic adaptation is the profound rewiring of metabolism at cellular and

physiological levels[52,53]. This study was driven by the observation in C2C12 cells and primary myoblasts, that hypothermic adaptation enhanced the ability of myoblasts to differentiate into myotubes (as judged by the cellular imaging and protein levels of differentiation markers) (Fig. 1). We observed that the ability of hypothermic adaptation to promote muscle differentiation was significant in both 3-month-old and 18-month-old mice (Fig. 1h, j), suggesting that this effect was not significantly affected by aging at least up to

**Fig. 5 | Analysis of metabolism, proliferation and differentiation of skeletal muscle myoblasts overexpressing RBM3 at 37 °C. a** Line graph showing the mitochondrial OCR of C2C12 cells overexpressing RBM3-GFP and GFP (control) and respectively. X-axis represents time in minutes and the y-axis represents oxygen consumption rate in pMol/min. **b** Bar graph quantifying the basal respiration, maximum respiration, spare respiratory capacity and ATP-linked respiration of C2C12 cells overexpressing RBM3-GFP and GFP respectively. Y-axis represents oxygen consumption rate in pMol/Min ($n = 3$). Significance tested by two-tailed student's $t$ test (unpaired). **c** Heat map showing the intracellular levels of TCA metabolites in C2C12 cells overexpressing RBM3-GFP and GFP respectively ($n = 3$). **d** Box plot showing the intracellular levels of acetyl-CoA in C2C12 cells overexpressing RBM3-GFP and GFP respectively ($n = 3$). **e** EdU staining of C2C12 cells overexpressing RBM3-GFP and GFP respectively. Red color indicates EdU stained nucleus and blue color indicates DAPI staining. **f** Bar graph representing the quantitation of EdU staining where the y-axis represents the % of EdU positive nucleus (17 data points taken from 2 individual experiments). **g** Bar graph quantifying the % viability of C2C12 cells overexpressing RBM3-GFP and GFP respectively at 37 °C ($n = 3$). **h** Bar graph indicating cell proliferation of C2C12 cells

overexpressing RBM3-GFP and GFP respectively at 37 °C where the y-axis represents the total number of cells, and the x-axis represents time in days ($n = 3$). Significance tested by two-tailed student's $t$ test (paired). **i** Western blot showing protein levels of MHC after 5, 6 and 7 days of differentiation using C2C12 myoblasts overexpressing RBM3-GFP and GFP respectively. **j** Bar graph quantifying the western blot in (**i**), where the y-axis represents the intensity of MHC normalized to the total protein ($n = 3$). **k** C2C12 myoblasts overexpressing RBM3-GFP and GFP respectively, differentiated for 6 days and immunostained with MHC antibody. **l** Bar graph quantifying the myotube fusion index (48 individual data points from 3 individual experiments) and myotube diameter of cells shown in (**k**) (28 individual data points from 2 individual experiments). **m** Primary myoblasts (from 3-month-old mice) overexpressing RBM3-GFP and GFP respectively, differentiated for 4 days and immunostained with MHC antibody. **n** Bar graph showing the myotube fusion index and myotube diameter quantified from the images shown in m (17 individual data points from 2 individual experiments). Error bar in the bar graphs represent standard error of mean (SEM). Significance tested by two-tailed student's $t$ test (unpaired). *, **, *** $p < 0.05$, 0.01, and 0.001 respectively.

18 months. This implies that hypothermic pre-conditioning of muscle myoblasts could be used to improve the differentiation of these cells even from older mice. We undertook a proteomics-based approach to map the cold-shock proteome using skeletal muscle myoblasts with the aim of identifying the underlying pathways that might play a role in adaptation to hypothermia. We subjected C2C12 mouse myoblasts to acute (6 h) and chronic (48 h) exposure to 32 °C. By analyzing this comprehensive dataset of the cold-shock proteome we observed that about 26% of the proteins detected were increased in levels, and 70% of proteins decreased in levels, as compared to the myoblasts incubated at 37 °C. This suggested that a significant percentage of the proteome responded to hypothermic stress. Notably, the GO pathway enrichment analysis and protein-protein network analysis revealed that there is a significant increase in levels of multiple structural histone proteins at 6 h of hypothermic stress (Fig. 2). Concomitantly pathways involved in prophase chromosome condensation were also enriched. C2C12 myoblasts under hypothermia were observed to have a slow proliferation rate and together this might suggest accumulation of myoblasts in prophase. The protein-protein network analysis underlines the role of the IL-17 signaling network in cellular response to hypothermia. It has been shown that IL-17 is increased in levels in response to injury or exercise and plays a role in muscle stem cell activity[54,55]. Skeletal muscle along with other metabolic tissue express the IL-17 receptor and this cytokine might act as an acute autocrine signal. Among proteins that were significantly down-regulated at 6 h of hypothermic stress, were those related to RNA metabolism and processing, and long-chain unsaturated fatty acid synthesis (Fig. 2c, d). This underlines the importance of post-transcriptional RNA regulation in cold response in agreement with previously reported studies where RBM3 has been shown to regulate or bind to proteins involved in splicing[31] and translation[47,56]. The cold-responsive RNA binding protein, RBM3[16], was also shown to be upregulated at 6 h (Fig. 4a). Interestingly, proteins related to unsaturated fatty acid biosynthesis are decreased in levels. (Fig. 2c, d). This suggests that membrane properties regulated via fatty acid desaturation may be an important target of hypothermic adaptation at earlier time points. We also analyzed proteins regulated by chronic (48-h) exposure of myoblasts to hypothermia and observed that about 26% of the total proteins detected were upregulated, again suggesting that a significant fraction of the proteome responds to hypothermia and could be responsible for phenotypes associated with adaptation to this stress. The most significant feature that we focused on in this study was the enrichment of pathways associated with cellular metabolism (Fig. 3). This suggests that metabolism is an adaptive response to hypothermia and might underlie the benefits of hypothermic pre-conditioning in muscle cells. Unlike the acute response, there is a significant upregulation of proteins involved in lipid metabolism, fatty acid biosynthesis, and peroxisomal transport (ACOX3, PTGR3, DHRS4). ACOX3 is known to be involved in the oxidation of lipids in peroxisomes[57–59]. Also, there was an overall upregulation in proteins

associated with monocarboxylic, aldehyde, glutathione metabolism (GSTM1, GSTA4, GSTM5), and RNA processing (RPP38, MRPS15, RTCA) (Fig. 3a, b). This suggests that cells might increase redox and carbon metabolism upon exposure to hypothermia. We noted that one of the factors that increased in levels both acutely and chronically was the RNA binding protein RBM3 (Fig. 4a). We tested the hypothesis that the maintenance of mitochondrial respiration in muscle myoblasts upon hypothermic pre-conditioning might be connected to the increase in levels of the cold-shock protein RBM3. RBM3 is highly expressed in long-lived strains of mice and has been shown to increase the survival of C2C12 cells under ROS and ER stress[60,61]. Using siRNA-based knockdown of RBM3, we showed that RBM3 is required to maintain mitochondrial respiration upon hypothermic pre-conditioning (Fig. 4f). We also showed that RBM3 is required to improve differentiation upon hypothermic pre-conditioning (Fig. 4d, e). This raised the possibility that the upregulation of RBM3 expression itself might be sufficient to recapitulate some of the effects of hypothermic adaptation on myoblasts. RBM3 has been shown to have cell protective effects in neuronal cells[36] and suggested to regulate translation of mRNA[56], though these mechanisms of action in multiple cellular contexts remain unclear. It has been shown to bind to polysomes and bind to other RNA binding proteins such as IGFBP2 and regulate its function[33]. Therefore, we explored the effects of the overexpression of RBM3 in myoblasts on cellular viability, proliferation, metabolism and differentiation. Two isoforms of RBM3 have been reported in primary astrocytes and hippocampal neurons and in neuronal cell lines such as B104 and N2A. These differ by the presence of an arginine residue (Arg135) in the C-terminal RG-domain[47]. The differential localization and expression of the RBM3 splice variant lacking this arginine residue (isoform 2) has been shown in the context of astroglia cells and neuronal dendrites[47]. In our study, we have used RBM3 isoform 2 as its cytoprotective effects have been widely implicated in cellular stress response.

We observed that mitochondrial respiration of C2C12 myoblasts overexpressing RBM3 (basal, maximum, ATP-linked respiration and spare respiratory capacity) (Fig. 5a, b) were significantly increased as compared to the GFP controls. Overexpression of RBM3 also caused resistance (as judged by the increase in IC50 concentrations) to the mitochondrial inhibitor oligomycin and the glycolytic inhibitor 2DG respectively (Fig. S3d, e). Finally, the overexpression of RBM3 also significantly increased intracellular levels of TCA cycle (Fig. 5c) metabolites and of Acetyl-CoA (Fig. 5d). Together this shows that RBM3 overexpression is sufficient to improve mitochondrial metabolism in C2C12 myoblasts. We next tested whether the improved metabolic phenotypes could affect proliferation and differentiation of both primary and C2C12 myoblasts. In agreement with this we observed that overexpression of RBM3 showed higher levels of EdU labeling, cell counts and MTT absorbance levels (Fig.5e, f, g, h). We assayed the levels of two metabolic enzymes that play a critical role in carbon

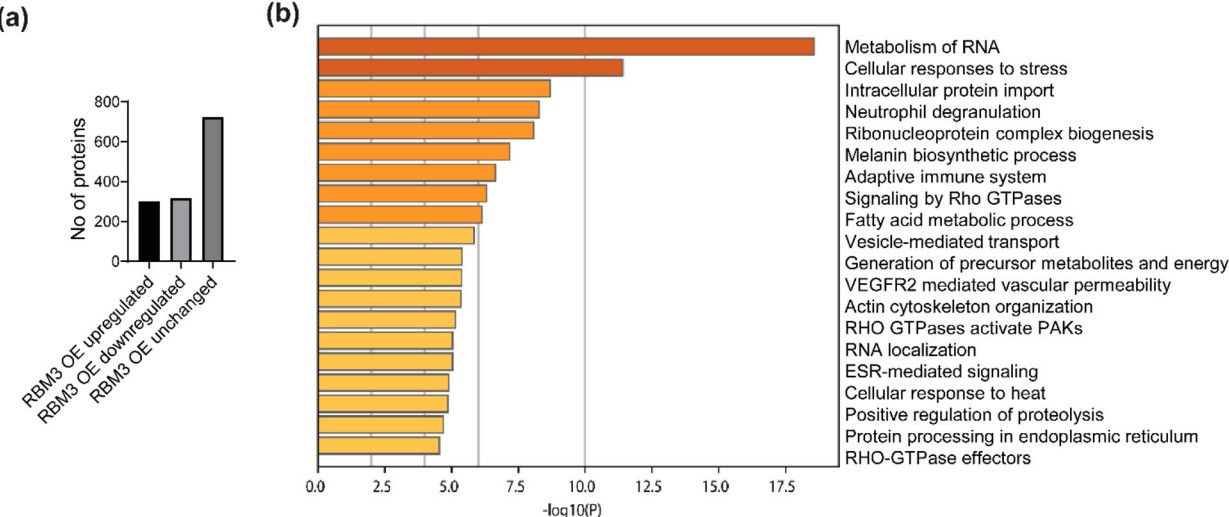

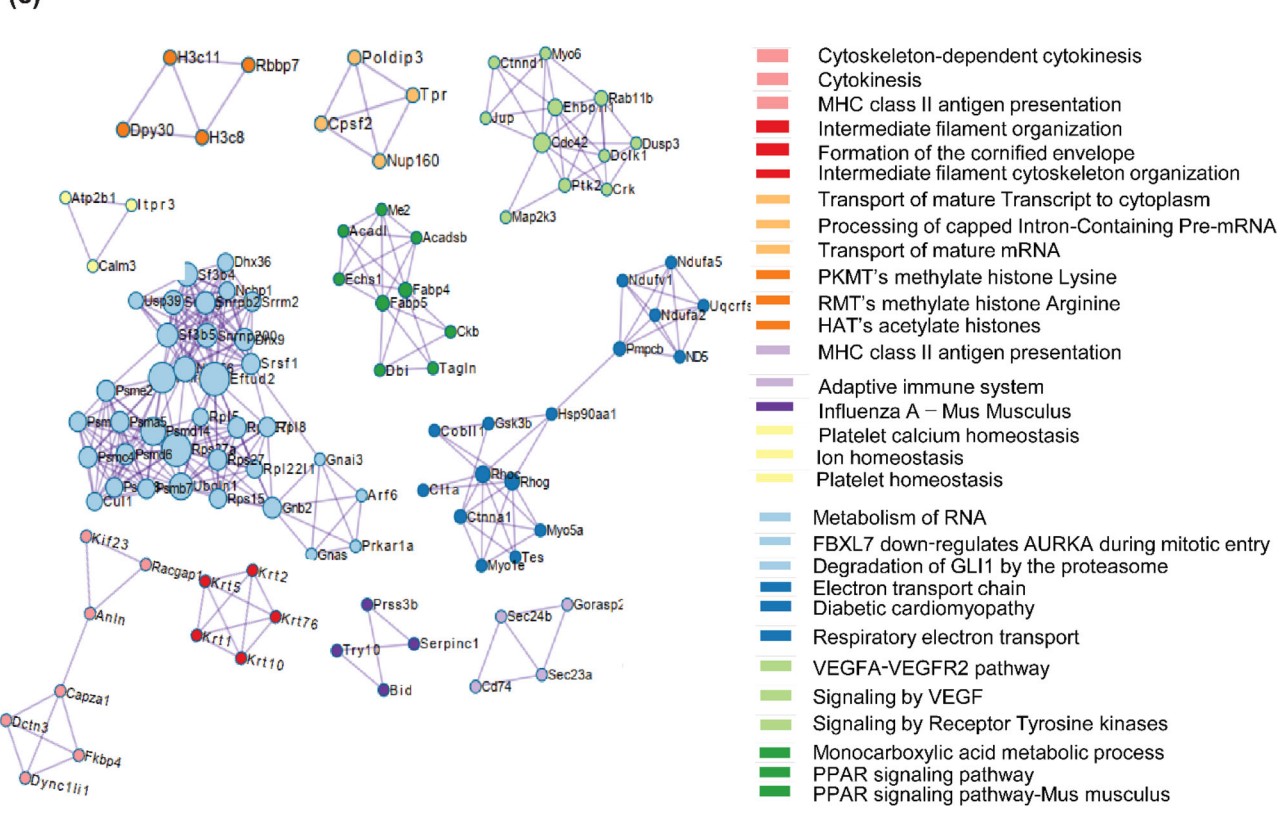

**Fig. 6 | Proteomic analysis of C2C12 myoblasts overexpressing RBM3 at 37°C.**
**a** Bar graph showing the number of upregulated, downregulated and unchanged proteins in C2C12 myoblasts overexpressing RBM3. **b** GO enrichment pathway analysis of upregulated proteins ($n = 3$). **c** GO enrichment analysis of protein-protein interaction networks of upregulated proteins ($n = 3$).

metabolism. The first was pyruvate dehydrogenase (PDH) which is the first step in the entry of pyruvate into the mitochondria as part of the TCA cycle[62]. By western blot analysis, we observed a significantly higher levels of PDH in C2C12 myoblasts overexpressing RBM3 compared to the GFP controls (Fig. S4d). Another metabolic enzyme pyruvate kinase (PKM) exist in two spliced isoforms (PKM1 and PKM2) and they control the metabolism and proliferation of numerous cell types[63,64]. The PKM2 to PKM1 ratio is increased in proliferative cells and has been shown to alter carbon flux into the mitochondria[65,66]. By western blot analysis we observed a significantly

higher levels in the ratio of PKM2 to PKM1 in myoblasts overexpressing RBM3 (Fig. S4e). Finally, we observed that the overexpression of RBM3 improved the levels of MHC protein in differentiated C2C12 myotubes (Fig. 5i, j). Immunofluorescence analysis showed that the fusion index and myotube diameter of C2C12 myotubes overexpressing RBM3 were higher than control, as another measure for improved differentiation (Fig. 5k, l). When primary myoblasts extracted from 3-month-old mice were differentiated into myotubes, we observed that the diameter of myotubes was significantly enhanced (but not the fusion index) (Fig. 5m, n) suggesting that

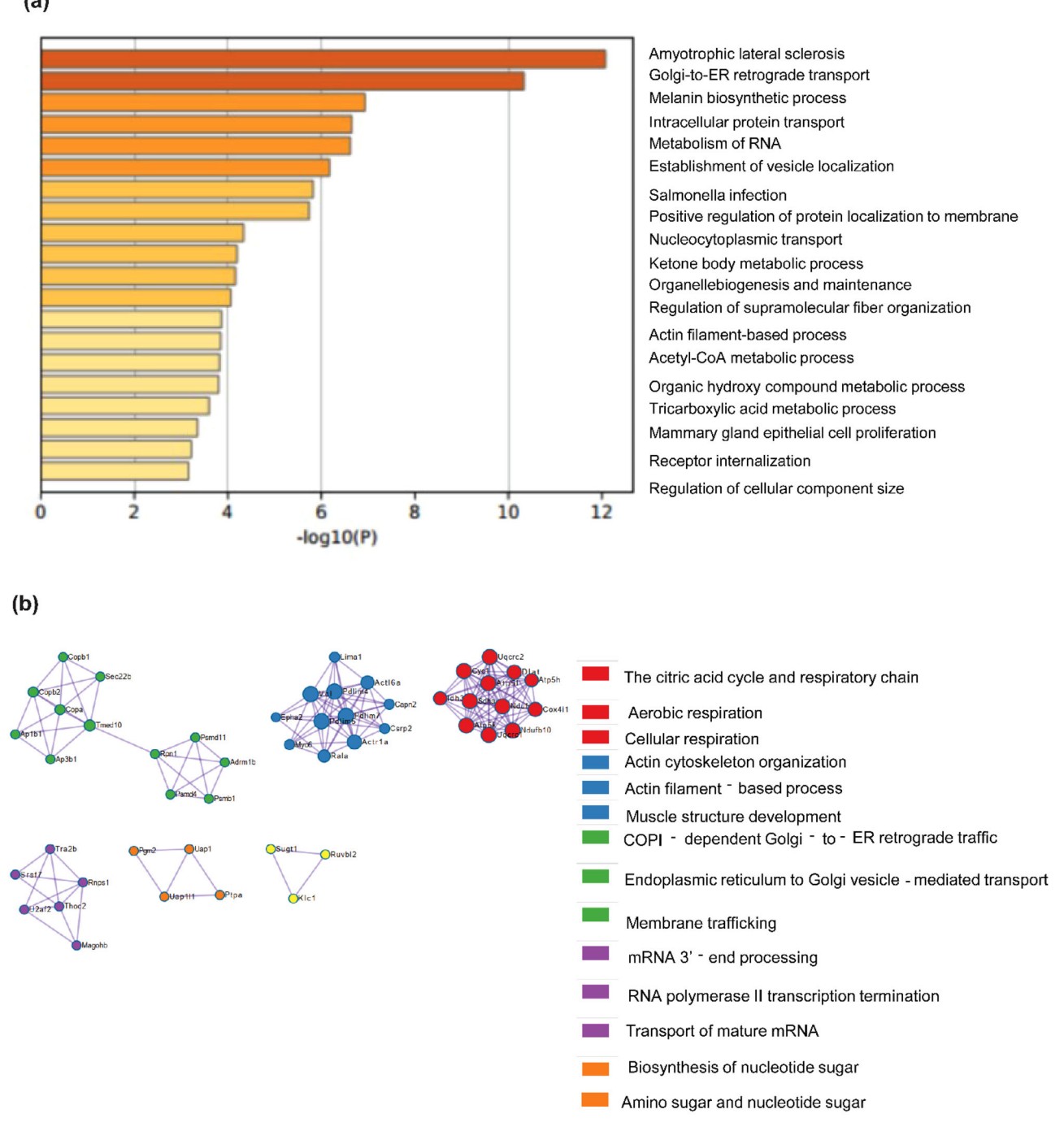

**Fig. 7 | Proteomic analysis of C2C12 myoblasts overexpressing RBM3 at 37°C. a** GO enrichment pathway analysis of downregulated proteins ($n = 3$). **b** GO enrichment analysis of protein-protein interaction networks of downregulated proteins ($n = 3$).

overexpression of RBM3 may improve the metabolism, leading to increased size of fibers in primary myotubes. To further elucidate the effects of RBM3 on skeletal muscle myoblasts we performed proteomics of C2C12 myoblasts over-expressing RBM3 or GFP respectively.

From the proteomics analysis, we observed an increase in proteins involved in PPAR signaling pathway (ACADL, ACADSB, ECHS1, DBI) in myoblasts overexpressing RBM3 compared to the control (Fig. S6b, c). ACADL, ACADSB, ECHS1 and DBI are enzymes involved in the beta oxidation of fatty acids[67,68]. This suggests that RBM3 might positively affect the beta oxidation of fatty acids in skeletal myoblasts. The

mitochondrial ETC proteins (NDUFA2, NDUFA5 and NDUFV1) (Fig. S6c) also increased in myoblasts overexpressing RBM3. NDUFs are mitochondrial complex I proteins which are involved in mitochondrial respiration[69–71]. This suggests that an enhanced mitochondrial respiration in cells overexpressing RBM3 could be due to an increase in the levels of proteins involved in the mitochondrial electron transport chain. It is notable that among proteins that were downregulated, numerous TCA associated enzymes including IDH1 and SDH. Since TCA enzymes are the source of NADH in the mitochondria, the down regulation (in the context of increase in levels of fatty acid oxidation enzymes) might

imply that cells rewire their metabolism to generated NADH from fatty acid oxidation instead of TCA.

In this study, we have shown that RBM3 directly controls the metabolism of muscle stem cells and their regenerative capacity in vitro. In the future, it will be important to determine the effect of RBM3 overexpressing myoblasts during skeletal muscle regeneration in vivo. The direct targets of RBM3 (both mRNA and protein) that mediate these effects need to be determined in the context of candidate-downstream effectors (e.g., BDNF-CREB axis[38], IGFBP2[33] and RTN3[32]). Controlling the expression of RBM3 under physiological conditions or using small molecule activators of RBM3 may present important avenues for regulating the skeletal muscle stem cell state and metabolism.

## Methods

### Cell culture techniques

C2C12 cells (purchased from ATCC) were grown and maintained in DMEM (Dulbecco's Modified Eagle Medium, Gibco-11995065) with 10% FBS (Foetal Bovine Serum, Gibco-16000044) and 1% Pen-Strep (Penicillin-Streptomycin, Gibco-15140122). Mouse primary myoblasts were cultured on ECM gel-coated dishes (Sigma-Aldrich-E6909). The cells were grown at 37 °C and 5% $CO_2$ conditions. C2C12 differentiation was done using DMEM + 2% horse serum (Gibco-26050070) and 1% Pen-Strep. Differentiation media was added once the cells reached 100% confluency and differentiation was done for 7 days. HEK293T cells were grown and maintained in DMEM with 10% FBS and 1% Pen-Strep. For hypothermic stress, cells were grown at 32 °C and 37 °C and 5% $CO_2$ conditions for 72 h.

### Preparation of retrovirus and stable cell lines

For retrovirus production, HEK293T was co-transfected with the plasmid containing the gene of interest pMIG-GFP, pMIG-RBM3 and packaging plasmid pCL-ECO using jetPRIME reagent (Polyplus-101000046). Media was collected 48 h and 72 h after transfection. The collected media was centrifuged at 4000 RPM for 5 min at room temperature. The supernatant was collected and concentrated using Retro-X-Concentrator from Takara Biosciences (Takara #631455) overnight at 4 °C. The following day the concentrate was centrifuged at 1500 RPM for 45 min at 4 °C to obtain the virus pellet. The pellet was resuspended in DMEM media and aliquoted and stored at −80 °C if not used immediately.

The retroviral titer calculation was done using the Retroviral titration kit from Takara Biosciences (Retro-X™ qRT-PCR Titration Kit 631453) using the qRT-PCR-based method.

C2C12 cells were transduced with $10^5 - 10^6$ virus particles. C2C12 cells were seeded in a 6 cm dish and $10^5 - 10^6$ virus particles were added dropwise to media containing 8 µg/mL of Sequabrene (Sigma S2667-1VL) and 25 mM Hepes buffer. The media was changed 24 h after transduction. Cells were checked 48 h after transduction and an efficiency of 90–95% was observed. All the experiments involving C2C12 cells overexpressing RBM3 were done by retro-viral transduction of C2C12 cells.

### si-RNA treatment on C2C12 cells

C2C12 cells were transfected with 200 nM of siRBM3 (Dharmacon L-041823-01-0005) and 200 nM scrambled control (Scr) (Dharmacon D-001810-10-05) using dharmafect transfection reagent following dharmafect transfection protocol.

### Protein isolation and western blotting

Protein isolation was done using RIPA buffer (Thermo Scientific-89901) and 1X PIC (Protease inhibitor cocktail cOmplete Roche 11697498001). The concentration of the protein was done using Bradford assay (Puregene PG-035-500ml) or BCA assay (G-Biosciences-786570). For western blotting 30ug protein was loaded in each well. 3% BSA in TBST was used as a blocking buffer. Primary antibodies: MHC (Invitrogen 14650382), RBM3 (Invitrogen PA5-51976), beta-ACTIN (CST 4967S), PKM1(CST D30G6), PKM2 (CST D78A4), PDH (CST 3205), phospho-4EBP1 (CST2855), Total 4EBP1 (CST 9452), phospho-eIF4E (ab76256), Total-eIF4E (ab1126) were

used in 1:1000 dilution and incubated at 4 °C overnight. Secondary antibodies: Anti-mouse IgG, HRP-linked (CST 7076) and Antirabbit IgG, HRP-linked (CST 7074) were used in 1:5000 dilution and incubated at room temperature for 1 h. The blots were developed using a developing solution (advansta-K-12049-D50).

### MTT assay

For the MTT assay, 5 mg/mL MTT reagent was used (Sigma M2128-500MG). Cells were seeded in a 96-well plate. For drug assay, 10000 cells were seeded per well. The following day, cells were treated with drugs oligomycin (Sigma 75351) and 2DG (Sigma D8375) at different concentrations: 50 nM, 100 nM, and 200 nM for oligomycin and 0.25 mM, 0.5 mM, 1 mM for 2DG for 48 h. After which cells were incubated with MTT reagent for 3 h at 37 °C in dark. Absorbance reading was taken at 570 nm using a plate reader. For the cell viability experiment, C2C12 stable lines: GFP and RBM3 have been seeded in 96 well plates 10,000 cells per well. MTT assay was done 48 h after seeding and absorbance was measured at 570 nm.

### Seahorse assay

Seahorse assay was done to measure the oxygen consumption rate (OCR) of the cells using Seahorse XFe24 bioanalyser machine. For that, the 24 well cell culture plates (Agilent Technologies 102340-100) were seeded with 18000 cells per well. DMEM XF base media (Agilent 102353-100) containing 1 mM sodium pyruvate (Sigma S8636-100ML), 2 mM glutamine (Sigma G7513-100ML) and 20 mM glucose (Qualigens Q15405) for measuring OCR was used. Cells were incubated in a non-CO2 incubator at 37 °C for 45 min. OCR was measured in response to the following drugs: 1uM oligomycin, 3uM FCCP (Cayman chemical 800364-9897), 1.5uM of Antimycin (Cayman chemicals 1397-94-0) and Rotenone (Sigma R8875-1G) for OCR. Apart from the drug treatment, OCR was measured at the baseline levels.

### Proteomics sample extraction and preparation

C2C12 cells were subjected to hypothermic conditions: 32 °C and 5% $CO_2$ for different time points of 6 h and 48 h. C2C12 stable cell lines overexpressing GFP and RBM3-GFP were used for proteomics studies. Cells were lysed with RIPA + Protease inhibitor cocktail. They were centrifuged at 10,000 rpm for 30 min at 4 °C. The supernatant was collected and a BCA assay was done. 100ug of protein was used for proteomic analysis. Chilled acetone was added to the protein samples and incubated for 3 h at −80 °C. Centrifugation was done at 13,000 rpm for 30 min at 37 °C. The pellet was washed with acetone at 13,000 RPM for 10 min at 37 °C. The samples were treated with 8 M urea (in Ammonium bicarbonate) and 0.5 M DTT and incubated for 30 min at 37 °C. Following this, the samples were treated with 30 mM iodoacetamide and incubated for 30 min at 37 °C. Equal concentrations of protein samples were digested using mass spectrometry grade trypsin (1:20) and digested peptides were desalted using a C18 SPE cartridge. Peptides were analyzed using a Sciex Zeno TOF 7600 (AB Sciex, Foster City, CA, United States) equipped with a Shimadzu LC 40 UPLC system. Peptides were separated on Aquity UPLC BEH C18 column (150 × 1 mm, 1.7 µm,) column using a gradient of water and Acetonitrile. For protein quantitation, spectral libraries were generated using information-dependent acquisition (IDA) mode after injecting 3 µg of tryptic digest on the above-mentioned C18 column using a Shimadzu LC40 autosampler system coupled with Sciex Zeno Tof 7600 fitted. For library generation, the peptides from all the experiment sets were pooled and injected into the mass spec for comprehensive coverage. The total chromatography run time was 180 min. The first 10 min of the column flow was sent to waste after loading for online desalting of the sample. A 180 min gradient in multiple steps (ranging from 5 to 30% acetonitrile in water containing 0.1% formic acid) was set up to elute the peptides from the column for the first 135 min then in the next 10 min concentration of B was increased to 55% and most of the peptide eluted till this time of 145 min. The concentration of mobile phase B was increased to 95% B in 6 min and held at this concentration for 5 min then the

column was again kept at 98% of mobile phase A (Water and 0.1% formic acid) for 24 min for equilibration. The flow rate of the UPLC was set at 30 μL min. Three biological replicates of the samples were run for each experimental set. For library preparation, the mass spec was run in an information-dependent acquisition mode (IDA) with the MS1 mass range of 350 Da to 1500 Da, Gas 1 was 25 L/min, Gas2 was set at 15 L/min, whereas curtain gas was set at 25 L/min, ionization voltage was 5500volt, accumulation time was set at 0.2 s and source temperature was at 350 °C. The most abundant top 50 multiple charge precursor was fragmented in each cycle with dynamic collision energy in collision-induced dissociation (CID). The mass range for MS/MS was set at 150 to 1800 Da. Zeno trapping was on for the IDA data.

## Sequential Window Acquisition of all Theoretical Fragment Ion Spectra (SWATH) analysis for label-free quantification

For label-free quantification (SWATH analysis) of cells overexpressing RBM3-GFP and GFP control, the Q1 transmission windows were set to 25 Da from the mass range of 350 Da to 1500 Da. A total of 79 windows were acquired independently with an accumulation time of 50 ms. The total cycle time was kept constant at 2 s. Protein Pilot$^{TM}$ v. 5.0.2 was used to generate the spectral library. For label-free quantification, peak extraction and spectral alignment were performed using PeakView$^{R}$ 2.2.0.11391 Software with the parameters set as follows: number of peptides, 2; number of transitions, 5; peptide confidence, 95%; XIC width, 20 ppm; XIC extraction window, 5 min. The data were further processed in MarkerView software $^{TM}$. 1.3.1 (AB Sciex, Foster City, CA, United States) for statistical data interpretation. In MarkerView$^{TM}$, the peak areas under the curve (AUC) for the selected transition were normalized using total area sum intensities and all the biological replicates were averaged out before normalization. $t$-test was performed on the data set and fold-changed values were calculated for all the proteins. Label-free (DIA) LCMS/MS analysis of 37 °C, 32 °C 6 h and 32 °C 48 h were acquired on Thermo LTQ Fusion. The digested peptides of the samples were analyzed using Themo LTQ Fusion platform using DIA workflow. The peptides were separated on C18 reverser phase column for 90 min and in DIA setup 12 Da overlapping windows were utilized in for data acquisition. The mass range and other parameters were kept similar to the other run on 7600 Zeno Tof. Data were analyzed using DIA-NN software.

## Data analysis for proteomics

Data were processed with Protein Pilot Software v. 5.0.2 (ABSciex, Foster City, CA, United States) utilizing the Paragon and Progroup Algorithm. The analysis was done using the tools integrated into Protein Pilot at a 1% false discovery rate (FDR) with statistical significance. In brief, the UniProt mouse proteome database (UP000000589) was used to search for the matched peptide for library generation. The download included total combined (reviewed and un-reviewed) entries of 55,286 proteins. The resultant search identification file was used as a library for the extraction of peptide quantitation information from the SWATH acquisition. The extracted peptide information was processed using Marker view software $^{TM}$(V1.3) for statistical analysis. Biological triplicate data for each sample were normalized using the total area sum method and all the biological replicates were averaged out for the calculation of fold change calculation. After normalization $t$-test and principal component analysis were performed on the data set to check the possible correlated variables within the group. A volcano plot was generated to calculate the statistically significant fold change vs $p$-value. Proteins with less than 0.6 fold change were considered as downregulated and proteins with more than 1.5 fold change were considered as up-regulated in the experiment sets. In this analysis library was made for approximately 2394 proteins that were identified with 0.05% FDR and ~1400 proteins were quantified in the data set. GO enrichment analysis, process and pathway enrichment analysis were done using Metascape software.

## Metabolite extraction and mass spectrometry

C2C12 myoblasts overexpressing RBM3-GFP and GFP were seeded in 10 cm cell culture dishes. Metabolite extraction was done using chilled 50% methanol. The cells were scrapped in 50% methanol and 100ug D4 alanine (Cambridge isotope lab DLM-250-1) and centrifuged at 13000 RPM for 5 min at room temperature. The supernatant was dried using a speed vacuum. The dried sample was stored at –80 °C. The pellet was resuspended in RIPA and protein was prepared and a BCA assay was done to determine the concentration. The media from the cells were processed similarly using 50% methanol. The dried samples were reconstituted and run on AB Sciex 5500 using the synergi rp fusion column.

For the detection of glycolytic metabolites and Acetyl-CoA, the samples were analyzed in a triple-quadrupole type mass spectrometer (Sciex QTRAP 5500) with Schmadzu HPLC unit. Multiple reaction monitoring (MRM) methods were modified as published by ref. 72. The metabolites were measured using Synergi, 4 μm Fusion-RP 80 A (150 × 2 mm; Phenomenex # 00F-4424-B0) by the methods described below: Samples were reconstituted in 50% methanol,100 μl Synergi, 4 μm Fusion-RP 80 A (150 × 2 mm; Phenomenex # 00F-4424-B0), Pump B; 0 min: 0%; 2.0 min: 5%; 6.7 min: 60%; 7.3 min: 95%; 9.3 min: 95%; 10 min: 5%; 10.3 min: 0%; 12 min: 0%; 12.1 min: controller stop; flow rate 0.2 ml/min; autosampler temperature 5 °C; Mobile phase A: 5 mM Ammonium Acetate (Sigma 73594-25G-F) pH 8.0; Mobile phaseB:100%acetonitrile (Fischer Scientific A955-4). The area under each peak was calculated using AB SCIEX MultiQuant software 3.0.1. For TCA intermediates along with lactate and pyruvate, the mobile phase consisted of a premixed ratio of Water/Methanol (95:5) with 0.2% formic acid. The column heating oven was set to 45 °C, autosampler temperature 5 °C; flow rate 0.4 ml/min. The MS method was used as such as per the manufacturer's instruction (Phenomenex TN-1241). Data were processed and analyzed using MetaboAnalyst software.

## Immunofluorescence and confocal microscopy

C2C12 and primary myoblasts (from 3-month-old mice) were seeded in coverslips and differentiated for 6 days at 37 °C after hypothermic pre-conditioning at 32 °C for 72 h. Cells were fixed with 4% PFA for 15 min at room temperature. Permeabilization was done using 0.1% triton-X-100 in 1X-PBS for 10 min at room temperature. Blocking was done for 1 hr using 5% BSA in 1X-PBST at room temperature. Cells were immunostained with anti-MHC antibody at a dilution of 1:300 and incubated at 4 °C over-night. and Alexa flour 568 goat anti mouse (Invitrogen A11031) was used at 1:500 dilution for 1 h at room temperature. For Nucleus staining, DAPI (Thermo Scientific 62248) was used at a dilution of 1:1000 for 5 min. Coverslips were mounted using Prolong gold mounting agent (Life Technologies P36930). Images of immunostained C2C12 were obtained using a confocal micro-scope. Images of immunostained primary muscle cells were obtained using an Epifluorescence microscope.

For Ki-67 immunostaining of C2C12 myoblasts overexpressing RBM3-GFP and GFP respectively, Cells were seeded on coverslips and after 12 h, they were shifted to media containing (DMEM + 0.1% FBS) and grown in serum-deprived media for 12 h. Cells were then fixed with 4% PFA for 15 min at room temperature. Permeabilization was done using 0.1% triton-X-100 in 1X-PBS for 10 min at room temperature. Blocking was done for 1 h using 5% BSA in 1X-PBST at room temperature. Cells were immunostained with anti-ki-67 antibody (Invitrogen PA5-19462) at a dilution of 1:300 and incubated at 4 °C overnight. The secondary antibody Alexa fluor 647 goat antirabbit (Invitrogen A21245) was used at 1:500 dilution for 1 h at room temperature. For Nucleus staining, DAPI (Thermo Scientific 62248) was used at a dilution of 1:1000 for 5 min. Coverslips were mounted using Prolong gold mounting agent (Life Technologies P36930). Images of immunostained C2C12 cells were taken using a confocal microscope.

## Puromycin labeling

SUnSET assay was used to measure in vitro protein synthesis in C2C12 myoblasts. C2C12 cells overexpressing RBM3-GFP and GFP were treated with 1 μg/ml puromycin for 30 min and 1 h at 80% confluency. Cells were collected in RIPA after 30 min and 1 h for protein isolation. The incorporation of puromycin in total protein was analyzed by western

blotting using anti-puromycin antibody (Millipore MABE343). This technique was adopted from ref. 73.

### EdU labeling
An equal number of C2C12 cells overexpressing RBM3-GFP and GFP were seeded on coverslips. After 24 h, cells were stained with EdU using Click-iT™ EdU Alexa Flour™ 647 Imaging kit (Invitrogen C10340).

### Satellite cell isolation and growth techniques
Satellite cells were isolated from 3 and 18-month-old BL6/J mice using the Miltenyi satellite cell isolation kit (Miltenyi biotech 130-104-268). Due to the low viability of cells using the kit-based method, a non-kit-based method was used to isolate satellite cells from aged mice[74]. The cells were grown in cell culture dishes coated with ECM (Sigma E6909) at 37 °C and 5% $CO_2$ conditions. Cells were cultured using growth media containing DMEM + 20% FBS + 1% penicillin–streptomycin. For differentiation of the primary cells, media containing DMEM + 5% horse serum + 1% penicillin-streptomycin was used. Primary cells were differentiated for 4 days. For hypothermic stress, cells were grown in 32 °C and 5% $CO_2$ conditions for 72 h. The Mice used for satellite cell isolation were housed at BLiSC Animal Care and Resource Centre (ACRC). The experiments were approved by the Institutional Animal Ethics Committee (IAEC) DBT-inStem (Institute for Stem Cell Science & Regenerative Medicine).

### Neon transfection
Primary mouse myoblasts were transfected with pMIG-RBM3 and pMIG-GFP using the Neon TM transfection system (Invitrogen MPK 1025). The cells were trypsinized and counted using a haemocytometer. For 12 well plates 3,00,000 cells and 2 µg DNA was used for transfection. The cell pellet was resuspended in 10 µl Buffer R and DNA was added to it. The cell suspension with DNA was placed on the neon tube containing 3 ml of Buffer E and the transfection program was run. The parameters used for transfection are voltage: 1500 V, pulse width: 10 ms, no of pulse: 3. The cells were grown without antibiotics for 48 h after transfection. After 48 h cells were grown in normal growth media containing antibiotics. All the experiments of mice primary muscle cells were done by electroporating the cells with pMIG-RBM3 and pMIG-GFP plasmids.

### Cloning techniques
Mouse RBM3 gene from RBM3-PUC plasmid (Origene MC203679) was cloned into pMIG-GFP plasmid by restriction digestion method using Bgl2 and EcoR1. Primers used for cloning (Forward primer: GGAA-GATCTTATGTCGTCTGAAGAAGGGAAACTC, Reverse primer: CCGGAATTCTCAGTTGTCATAATTGTCTCT). The clone sequences were verified using the Sanger sequencing technique. All the plasmids were purified using Qiagen miniprep kit (Qiagen 27104). For neon transfection of primary myoblasts, pMIG-RBM3-GFP plasmid was used.

### Statistics and reproducibility
All the data are represented as mean ± SEM. All the bar graphs were generated using graph pad Prism 8 using unpaired student's $t$-test, paired student's $t$-test or two-way ANOVA using multiple comparisons as mentioned in the respective figure legends. Statistical significance was accepted at $p \leq 0.05$

### Data availability
All the mass spectrometry and proteomics raw and processed data files are available on figshare for review and accessibility with the following doi: 10.6084/m9.figshare.25449259. Source data behind the graphs and charts in the paper can be found in Supplementary Data 1. All the uncropped original western blot images have been provided in the supplementary file as Fig. S8.

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

## Acknowledgements

The authors would like to acknowledge funding from the Department of Biotechnology, Govt of India. PD was funded by the CSIR fellowship. We thank Prof. Apurva Sarin for pMIG-GFP plasmid.

## Author contributions

P.D. contributed to the cell culture, molecular biology, biochemical assays and writing of the manuscript. S.R. contributed to the cell culture and molecular biology work. P.S.T. contributed to the cell culture and molecular biology work. N.Sa contributed to western blot analysis. P.S. contributed to western blot analysis and epifluorescence image analysis. M.A.H. performed metabolomic analysis. H.L. performed confocal imaging and satellite cell isolation experiments. N.S. contributed to the proteomic studies. A.R. participated in the conceptualization and writing of the manuscript.

## Competing interests

All authors declare no competing interests.
