## [Peer Review File · Communications Biology]

Reviewers' comments:

Reviewer #1 (Remarks to the Author):

General Comments

The manuscript describes the investigation into the role that cold shock protein RBM3 plays in the function of skeletal muscle myoblasts. All experiments are performed in vitro in C2C12 muscle cells or in primary mouse myoblasts. Hypothermia is known to exert protective effects on muscle cells and this paper proposes that the effect is due to the actions of RBM3. The significance of the experiments is high because it potentially points towards an intervention that could aid in muscle differentiation, particularly in aged. Also, since RBM3 is expressed in all cells, the protective effect could be translated to other cells and other field of study could benefit. In general, the experiments are well performed and consistent with showing the effect of RBM3 and hypothermia. However, there are some major issues associated with the presentation of the data, the description of the results and the interpretation of the data that need to be addressed. The Discussion seems to be merely a restatement of results and should be rewritten.

Major comments:

All figures should be shown with individual data points printed in the graphs for transparency. In addition, the legends are too long and need to be substantially condensed.

In most graphs no statistical significance is shown with signs even though in the legends the signs are explained. It is therefore not clear what is significantly different and what is not. The results description is therefore hard to follow since there is so little significance shown in the graphs.

In the entire manuscript the words 'increase' and 'decrease' need to be changed to 'higher than..' and 'lower than..', since the comparisons are not in the same cells and therefore only differences and no changes can be inferred. The same is true for 'upregulation'. Also, the word 'changed' should be replaced by 'difference' in the text and in all figures. This is extremely important, because in a lot of places in the Results description it is hard to follow what is different from what.

In general, in the entire document it needs to be better explained which 'cells' are meant when the word is used. Sometimes it is myoblasts and sometimes it is myotubes. This needs to be better described.

The authors missed one important manuscript in the literature: Smart et al., 2007, J Neurochem., 101, 1367-1379. This manuscript describes not only the fact that there are two RBM3 isoforms, but also that RBM3 induces proteins involved in protein translation. Therefore, the authors need to describe in their methods which of the two isoforms they used for overexpression and cite this paper when discussing translation factors.

The age of the mice is not very informative for the experiments. The young ones are very young and 18 months is not old for these mice. The conclusions need to reflect this fact.

There is some mention of n=1. No statistics can be performed on n=1. More experimental data need to be added.

The Discussion is merely a restatement of results and very little context is discussed. The significance of the findings is not placed in a broader picture of the function of RBM3 in general and in muscle in particular. Some of the proposed functions are over-reaching and too speculative and this needs to be changed. Also, there are too many paragraphs and they do not follow general rules for writing paragraphs. In this reviewer's opinion, the Discussion needs to be rewritten.

In the Figures there should be representative pictures of the cells shown.

Minor comments

Introduction:

1. Add the word 'skeletal' before myoblasts in line 35.
2. Line 37: Reference 8 is not correct in this context. The paper does not describe cell replacement therapies. This reference needs to be corrected or the statement needs to be removed.
3. Line 49: this statement requires a reference.
4. Line 62: the word 'rejuvenate' should be removed. These cells are not rejuvenated. They only increase some stem cell markers, which does not imply rejuvenation.

Results

1. Line 81: 'Cells' should be explained better: myoblasts or myotubes?
2. Line 189: What does the term 'systemic rewiring' mean? This is very vague and really does not describe anything biological unless more context is provided.
3. Line 278: protein levels were not quantified: this needs to be done and reported.

References

1. There are quite a few mistakes in the references. This needs to be fixed.

Methods:

1. The word 'mass spec' needs to be written out in full.

Statistics

1. For the time course experiments a repeated ANOVA needs to be performed. Also, the post hoc tests for the ANOVAS need to be mentioned.

Figures

1. Why is d4 not shown in Figure 4 F & G?

Reviewer #2 (Remarks to the Author):

In this report, Dey et al. investigated the cellular response to hypothermia in myoblasts. Primary findings were that subsequent differentiation is improved in myoblasts exposed to cold, and this effect appears to be mediated at least in part by RBM3, a known cold-induced protein. Indeed, RBM3 overexpression recapitulated many of the effects of cold exposure in myoblasts. The authors show that the effects of RBM3 expression may be mediated by increased mitochondrial respiration and regulation of metabolism, PKM2, and regulation of the AMPK/mTOR axis. This is an important area of research, both in the context of cryotherapy but also, perhaps, in pharmacological approaches for old age and other conditions in which muscle regeneration is compromised.

The work is novel for the most part. It extends previous findings in the literature that RBM3 is induced by cold in muscle cells; the proteomics extend and, for the most part, agree with previously published transcriptomic work and is consistent with previous findings that RBM3 limits apoptosis and promotes survivability and growth signaling. The primary novelty of this study is showing the effect of cold and RBM3 on muscle differentiation.

I have the following comments and suggestions:

- 1) General: Throughout the paper, differences in "differentiation" are assumed based on RNA or protein expression of MHC or other myogenic markers. The addition of a more direct and convincing morphological marker of differentiation would be very helpful (e.g., differentiation indexing of MHC-stained myotubes).
- 2) Line 49. The assertion that RBM3 can inhibit atrophy needs a reference.
- 3) Figure 3: In terms of logical flow, I think figure 3 and 2 should be swapped. The results section describes the data in Figure 3 first, then Figure 2.
- 4) Line 128-129: This needs to be reconciled with Figure 2G, which shows no increase in RBM3 at 48

hours.

5) Statistics: The approach is fine by me, but only a few figures (e.g., 1B, 4I, 4J, 6E, 6F, Fig 7) have indicators of statistical significance. For instance, Line 72 indicates that myogenin and MHC protein increased with hypothermia, but Figures 1F and 1G have no significance indicators). This should be reconciled throughout the figures.

6) Figure 1F,G, H: Is there any reason why tubulin was used as the reference protein in F, but Actin in G and H? And based on the representative blots, I'm not sure those are good reference proteins for this study anyway – they seem quite variable. I would suggest using total protein (e.g., ponceau staining) or another approach to normalizing the blot data.

7) Line 239-256 and figure 8: The authors found no difference in phosphorylation of AMPKalpha at Thr172 (which is related to AMPK activity), but decreased AMPKbeta phosphorylation at Ser182 (which is not related to AMPK activity, but to its localization). How this change relates to the decreased ACC phosphorylation is not well described in the paper. It seems likely that the decreased AMPK phosphorylation at Ser182 would lead to increased localization of AMPK in the nucleus instead of the cytoplasm (Warden et al., *Biochem J.*, 354, 275-283, 2001). This would decrease its ability to phosphorylate cytoplasmic ACC, explaining the decreased ACC phosphorylation and suggesting that cytoplasmic AMPK activity is decreased by lower cytoplasmic AMPK content. However, this is quite speculative. Also, the authors measured 4EBP1 phosphorylation and suggested that increased 4EBP1 activity may be driving the increase in ribosomal proteins. However, the relationship between AMPK and mTORC1 signaling is poorly described here. It seems likely that decreased AMPK activity is at least partly contributing to the increased 4EBP1 phosphorylation and, presumably, protein synthesis. This storyline seems a little tenuous to me but clearly tying together the potential mechanism here might help to clarify the relevance of the AMPK/4EBP1 findings to the overall storyline.

Reviewer #3 (Remarks to the Author):

Figure 1 and data therein:

The schematic in Figure 1A is somewhat unclear. What does the blue arrow represent? Why does it point right?

Throughout the text of the results section, there are statements about how temperature changes the levels of many of the transcripts or proteins shown. However, there are no corresponding statistical differences except for one of the five comparisons shown in 1B. One cannot simply decide that something is different without a statistical analysis demonstrating it.

In Figure 1, why would both mRNA and protein be measured and shown? The mRNA has no activity here; only the protein does. The mRNA is simply an intermediate. If the mRNA and protein data disagreed (as they often do), it would only make sense to focus on the protein data. For this reason, the mRNA data might be best included in the supplemental data section.

From lines 83 onward: what does the statement 'upregulated at 6h, 12h, etc' actually mean? Is this in comparison to a 0h measurement always? Or is it compared to the 37C group? Please clarify in the text and figures.

25C killed the primary myoblasts but not the C2C12 myoblasts. Why? This seems like an important result, but it is not presented.

Lines 81-82 and the following section. I am somewhat puzzled why much is made of the fact that metabolic proteins are increased in C2C12 cells cultured at 32C. This is a classic biochemical response: at the lower temperature, fewer reactant molecules possess sufficient kinetic energy to meet the threshold for an enzyme-catalyzed reaction. The cell responds by making more enzyme as a way to maintain the rate of the reaction. This has been demonstrated many times in the literature on cold acclimation from the 1980s to present.

Figure 3 and throughout the manuscript: continually, statements are made about upregulation, increases, etc, but rarely is there any statistical analysis to back it up. Or at least the symbols that would be used to show this are not present. This must be addressed BEFORE the story is reviewed. If many of the statements cannot be backed up with a mathematical demonstration of probability, then the statements cannot be made. This will change the entire text of the Results section.

This occurs again in the presentation of mitochondrial effects in Figure S7, where we are told that there was no significant difference in mitochondrial morphology or membrane potential, but just shown a single representative image of cells from each experimental group. The term 'significant difference' has a very specific meaning in biology. It is being misused here.

We are extremely grateful to the reviewers for their thoughtful feedback in strengthening the science and sharpening our focus. We have completely rewritten the manuscript and reordered figures as suggested and redone experiments to strengthen the statistical and other aspects of the manuscript throughout.

Thank you.

Reviewer #1 (Remarks to the Author):

General Comments

The manuscript describes the investigation into the role that cold shock protein RBM3 plays in the function of skeletal muscle myoblasts. All experiments are performed in vitro in C2C12 muscle cells or in primary mouse myoblasts. Hypothermia is known to exert protective effects on muscle cells and this paper proposes that the effect is due to the actions of RBM3. The significance of the experiments is high because it potentially points towards an intervention that could aid in muscle differentiation, particularly in aged. Also, since RBM3 is expressed in all cells, the protective effect could be translated to other cells and other field of study could benefit. In general, the experiments are well performed and consistent with showing the effect of RBM3 and hypothermia. However, there are some major issues associated with the presentation of the data, the description of the results and the interpretation of the data that need to be addressed. The Discussion seems to be merely a restatement of results and should be rewrite.

Major comments:

All figures should be shown with individual data points printed in the graphs for transparency. In addition, the legends are too long and need to be substantially condensed.

Response- Individual datapoints have been added and the legends have been condensed.

In most graphs no statistical significance is shown with signs even though in the legends the signs are explained. It is therefore not clear what is significantly different and what is not. The results description is therefore hard to follow since there is so little significance shown in the graphs.

Response- Clear statistical signs and tests have been added

In the entire manuscript the words 'increase' and 'decrease' need to be changed to 'higher than..' and 'lower than..', since the comparisons are not in the same cells and therefor only differences and no changes can be inferred. The same is true for 'upregulation'. Also, the word 'changed' should be replaced by 'difference' in the text and in all figures. This is extremely important, because in a lot of places in the Results description it is hard to follow what is different from what.

Response- We have made these change to wordings and providing clear text what samples are compared against in the manuscript in the document. In the proteomics section we have kept the term 'upregulated' in the context of pathways, but have removed this word in the rest of the manuscript.

In general, in the entire document it needs to better be explained which 'cells' are meant when the word is used. Sometimes it is myoblasts and sometimes it is myotubes. This needs to be better described.

Response- We have ceased using the word 'cells' and have specified myoblasts or myotubes throughout

The authors missed one important manuscript in the literature: Smart et al., 2007, J Neurochem., 101, 1367-1379. This manuscript describes not only the fact that there are two RBM3 isoforms, but also that RBM3 induces proteins involved in protein translation. Therefore, the authors need to describe in their methods which of the two isoforms they used for overexpression and cite this paper when discussing translation factors.

Response- We cite this paper and specify that we have used isoform 1 in our experiments based on precedence in the references.

The age of the mice is not very informative for the experiments. The young ones are very young and 18 months is not old for these mice. The conclusions need to reflect this fact.

Response- We have ceased referring mice as 'old' or 'young' and articulate the age specifically in the text.

There is some mention of n=1. No statistics can be performed on n=1. More experimental data need to be added.

Response- We have performed multiple replicates, and have provided the n's and statistical tests accordingly, including in the western blots where bands have been quantified and statistical test provided as required.

The Discussion is merely a restatement of results and very little context is discussed. The significance of the findings is not placed in a broader picture of the function of RBM3 in general and in muscle in particular. Some of the proposed functions are over-reaching and too speculative and this needs to be changed. Also, there are too many paragraphs and they do not follow general rules for writing paragraphs. In this reviewer's opinion, the Discussion needs to be rewritten.

Minor comments

Introduction:

1. Add the word 'skeletal' before myoblasts in line 35.
2. Line 37: Reference 8 is not correct in this context. The paper does not describe cell replacement therapies. This reference needs to be corrected or the statement needs to be removed.
3. Line 49: this statement requires a reference.
4. Line 62: the word 'rejuvenate' should be removed. These cells are not rejuvenated. They only increase some stem cell markers, which does not imply rejuvenation.

Results

1. Line 81: 'Cells' should be explained better: myoblasts or myotubes?
2. Line 189: What does the term 'systemic rewiring' mean? This is very vague and really does not describe anything biological unless more context is provided.
3. Line 278: protein levels were not quantified: this needs to be done and reported.

References

1. There are quite a few mistakes in the references. This needs to be fixed.

Methods:

1. The word 'mass spec' needs to be written out in full.

Statistics

1. For the time course experiments a repeated ANOVA needs to be performed. Also, the post hoc tests for the ANOVAS need to be mentioned.

Figures

1. Why is d4 not shown in Figure 4 F &G?

Response- We are extremely grateful for this detailed and thoughtful list. we have rewritten the entire manuscript and addressed these points.

Reviewer #2

In this report, Dey et al. investigated the cellular response to hypothermia in myoblasts. Primary findings were that subsequent differentiation is improved in myoblasts exposed to cold, and this effect appears to be mediated at least in part by RBM3, a known cold-induced protein. Indeed, RBM3 overexpression recapitulated many of the effects of cold exposure in myoblasts. The authors show that the effects of RBM3 expression may be mediated by increased mitochondrial respiration and regulation of metabolism, PKM2, and regulation of the AMPK/mTOR axis. This is an important area of research, both in the context of cryotherapy but also, perhaps, in pharmacological approaches for old age and other conditions in which muscle regeneration is compromised.

The work is novel for the most part. It extends previous findings in the literature that RBM3 is induced by cold in muscle cells; the proteomics extend and, for the most part, agree with previously published transcriptomic work and is consistent with previous findings that RBM3 limits apoptosis and promotes survivability and growth signaling. The primary novelty of this study is showing the effect of cold and RBM3 on muscle differentiation.

I have the following comments and suggestions:

1) **General:** Throughout the paper, differences in "differentiation" are assumed based on RNA or protein expression of MHC or other myogenic markers. The addition of a more direct and convincing morphological marker of differentiation would be very helpful (e.g., differentiation indexing of MHC-stained myotubes).

Response- we have rerun all experiments with western blots. We have also used immunohistochemical analysis and quantitated fusion index for the myotubes, and included representative images

2) **Line 49.** The assertion that RBM3 can inhibit atrophy needs a reference.

Response- We have provided the reference

3) **Figure 3:** In terms of logical flow, I think figure 3 and 2 should be swapped. The results section describes the data in Figure 3 first, then Figure 2.

Response- We have rewritten the manuscript in its entirety, and changed the flow as suggested

4) **Line 128-129:** This needs to be reconciled with Figure 2G, which shows no increase in RBM3 at 48 hours.

Response- We have rewritten the manuscript in its entirety, the figure numbers have changed, Now we have removed all the mRNA data and showed only protein data, which removes this discrepancy.

5) Statistics: The approach is fine by me, but only a few figures (e.g., 1B, 4I, 4J, 6E, 6F, Fig 7) have indicators of statistical significance. For instance, Line 72 indicates that myogenin and MHC protein increased with hypothermia, but Figures 1F and 1G have no significance indicators). This should be reconciled throughout the figures.

Response- We have rewritten the manuscript in its entirety, we have added statistical significance to all the figures.

6) Figure 1F,G, H: Is there any reason why tubulin was used as the reference protein in F, but Actin in G and H? And based on the representative blots, I'm not sure those are good reference proteins for this study anyway – they seem quite variable. I would suggest using total protein (e.g., ponceau staining) or another approach to normalizing the blot data.

Response- We have remade all figures. Now we now used Ponceau staining for all the gels and used total protein for quantitation instead of Actin or tubulin.

7) Line 239-256 and figure 8: The authors found no difference in phosphorylation of AMPKalpha at Thr172 (which is related to AMPK activity), but decreased AMPKbeta phosphorylation at Ser182 (which is not related to AMPK activity, but to its localization). How this change relates to the decreased ACC phosphorylation is not well described in the paper. It seems likely that the decreased AMPK phosphorylation at Ser182 would lead to increased localization of AMPK in the nucleus instead of the cytoplasm (Warden et al., *Biochem J.*, 354, 275-283, 2001). This would decrease its ability to phosphorylate cytoplasmic ACC, explaining the decreased ACC phosphorylation and suggesting that cytoplasmic AMPK activity is decreased by lower cytoplasmic AMPK content. However, this is quite speculative. Also, the authors measured 4EBP1 phosphorylation and suggested that increased 4EBP1 activity may be driving the increase in ribosomal proteins. However, the relationship between AMPK and mTORC1 signaling is poorly described here. It seems likely that decreased AMPK activity is at least partly contributing to the increased 4EBP1 phosphorylation and, presumably, protein synthesis. This storyline seems a little tenuous to me but clearly tying together the potential mechanism here might help to clarify the relevance of the AMPK/4EBP1 findings to the overall storyline.

Response- We thank the reviewer for these thoughtful mechanistic suggestions. We have tested the hypothesis that AMPK localization to the nucleus that might explain the effects of phosphorylation at Ser182. After extensive testing of the hypothesis using western blotting and microscopy we were unable to confirm the posited hypothesis, and agree with the reviewer that this aspect of the paper is unformed. Instead, we have tested whether total translation is increased in cell overexpressing RBM3 (new Figure S7) using puromycin staining. We still show the 4E-BP1 phosphorylation increase, but this does not imply that total translation is increased based on the puromycin data. We are unable to explain this based on the data we have. We think this is beyond the scope of the paper, and will pursue in the following manuscripts.

Reviewer #3:

Figure 1 and data therein:

The schematic in Figure 1A is somewhat unclear. What does the blue arrow represent? Why does it point right?

Response- we have removed the schematic as it was not helpful to the reviewer and caused more confusion.

Throughout the text of the results section, there are statements about how temperature changes the levels of many of the transcripts or proteins shown. However, there are no corresponding statistical differences except for one of the five comparisons shown in 1B. One cannot simply decide that something is different without a statistical analysis demonstrating it.

Response- We have rewritten the manuscript and solidified the statistical analysis and increased replicates wherever necessary.

In Figure 1, why would both mRNA and protein be measured and shown? The mRNA has no activity here; only the protein does. The mRNA is simply an intermediate. If the mRNA and protein data disagreed (as they often do), it would only make sense to focus on the protein data. For this reason, the mRNA data might be best included in the supplemental data section.

Response- we have rerun all experiments, and switched to only using protein data in our all experiments.

From lines 83 onward: what does the statement 'upregulated at 6h, 12h, etc' actually mean? Is this in comparison to a 0h measurement always? Or is it compared to the 37C group? Please clarify in the text and figures.

Response- We have simplified the manuscript and clarified wherever possible what the measurements are in relation to.

25C killed the primary myoblasts but not the C2C12 myoblasts. Why? This seems like an important result, but it is not presented?

Response- In the current form, we have narrowed our focus only to 32°C hypothermia, and no longer refer to this result as it does not add to the message in the paper any way.

Lines 81-82 and the following section. I am somewhat puzzled why much is made of the fact that metabolic proteins are increased in C2C12 cells cultured at 32C. This is a classic biochemical response: at the lower temperature, fewer reactant molecules possess sufficient kinetic energy to meet the threshold for an enzyme-catalyzed reaction. The cell responds by making more enzyme as a way to maintain the rate of the reaction. This has been demonstrated many times in the literature on cold acclimation from the 1980s to present.

Response- Thank you for raising this point. The hypothesis put forth by the reviewer is a good one but clearly not supported by the proteomics results. The interesting finding is that only a subset of metabolic proteins even in the same pathway are upregulated. In fact some are downregulated. What our paper reveals that the hypothermic response is much more nuanced. We propose that there is an actual rewiring that now prefers one set of pathways over another, which manifests in the preconditioning response. To our knowledge this is the first comprehensive and more unbiased study, using proteomics. In an unbiased manner a few important pathways (fatty acid pathways and not glycolysis), from metabolism to RNA processing, reveal themselves as more important in this rewiring. We are hard pressed to find such a systematic study like this, except for an exceptional study by Rabinowitz et. al; last year in Cell Metabolism (which we have cited) which shows such an approach albeit in a whole animal study.

Figure 3 and throughout the manuscript: continually, statements are made about upregulation, increases, etc, but rarely is there any statistical analysis to back it up. Or at least the symbols that would be used to show this are not present. This must be addressed BEFORE the story is reviewed. If many of the statements cannot be backed up with a mathematical demonstration of probability, then the statements cannot be made. This will change the entire text of the Results section.

Response- We have rewritten the manuscript and solidified the statistical analysis and increased replicates wherever necessary.

This occurs again in the presentation of mitochondrial effects in Figure S7, where we are told that there was no significant difference in mitochondrial morphology or membrane potential, but just shown a single representative image of cells from each experimental group. The term 'significant difference' has a very specific meaning in biology. It is being misused here.

Response- We have multiple replicates of this result. This did not add anything significant to the papers findings in our judgement and therefore it has been removed in this version of the manuscript.

Reviewers' comments:

Reviewer #1 (Remarks to the Author):

I appreciate the author's revision. They have addressed all of my concerns.

Reviewer #2 (Remarks to the Author):

The revised manuscript of Dey and al. has been already deeply modified and comments from the three reviewers have been taken into consideration.

The relevance of the data and the interest of the scientific community remains high, but the presentation can still be improved.

- Lines 140 and 144 : Figure 1 panels F, G,H and I are mislabeled related to the figure. (no G line 140, G and I line 144)

- Line 158-159 : there are no indications why the authors choose to select differentially regulated protein on fold change (0.6-1.5) rather than on adjusted p-value. Have all selected proteins a significant change ?

- Lines 174, 200-201, 248-249, and others. The authors misinterpreted the data, considering that if a set of proteins is present at a lower amount, then the corresponding pathway is downregulated. Similarly, if a set of proteins appears at a higher amount, then the corresponding pathway is upregulated. This can be often true, but not always, as some increased proteins can be inhibitors of the pathway and conversely. Only a careful examination of the considered set of proteins can lead to a putative statement on the respective pathway.

- Lines 221-222 : CIRBP is also increased at 12h and 24h

- Line 244: (fig4D,E) : there is a lonely star in the figure 4 panel E (comparison between D0 and D6 RBM3 siRNA)

- Lines 267-268 (fig5A-B): On panel A, red squares (maximum respiration in RBM-GFP conditions) are below the 150 pMol/min mark, whereas in panel B right, the corresponding bar indicates a mean between 150 and 200 pMol/min.

- Line 324: it is not clear from the text, legend and mat&meth section whether the cells were transfected or infected, depending on the use of plasmids or viruses to overexpress the recombinant protein.

- Lines 335-336: same as above. What is an increase in RNA metabolism, when it can be a change in RNA synthesis, steady state, or degradation? Only a careful examination of the protein list and their functions (biological processes) can help to hypothesise the biological consequences.

- Line 346 : why up- and down-regulated proteins are split into main and supplementary figures?

General comments :

- It would have been informative to compare lists of differentially expressed proteins in cold stress (myoblasts exposed to 32°C) and in RBM3 overexpressing myoblast. How the two lists overlap could help to better determine the role of RBM3 in cold stress response, and better characterize RDM3 direct targets.

- The discussion section is too long with too many repetitions of results.

- The experiments were mainly performed in myoblasts (cold stress and/or RBM3 under-/overexpression. As myoblasts only represent a tiny percentage of the skeletal muscle tissue, could the authors justify their choice to explore the impact of cold stress in myoblasts instead of myotubes?

Authors remark to Reviewer #2:

1. Lines 140 and 144 : Figure 1 panels F, G,H and I are mislabeled related to the figure. (no G line 140, G and I line 144).

Thank you for bringing this to our attention. The text in line 135-138, 140-142, and 143-145 has been revised accurately as per Figure 1 panel F, G, H, I, and J (corrections highlighted in yellow). Same has been corrected in figure legends (line 734-743).

2. Line 158-159 : there are no indications why the authors choose to select differentially regulated protein on fold change (0.6-1.5) rather than on adjusted p-value. Have all selected proteins a significant change ?

The data set generated from proteomic analysis was processed based on two-step filtering criteria. Only the first step was based on fold-change criteria (≥ 1.5 for upregulation and ≤ 0.6 for downregulation). In the second step we calculated statistical significance of the fold-change (line 154-157). However, not all proteins were significantly altered (p-values of all fold changes are now included in the supplementary data). Proteins with significant changes are illustrated in heat maps S2 A, B, C, and D. Significance levels are denoted by *, **, and ***, representing p values < 0.05 , 0.01 , and 0.001 , respectively.

3. Lines 174, 200-201, 248-249, and others. The authors misinterpreted the data, considering that if a set of proteins is present at a lower amount, then the corresponding pathway is downregulated. Similarly, if a set of proteins appears at a higher amount, then the corresponding pathway is upregulated. This can be often true, but not always, as some increased proteins can be inhibitors of the pathway and conversely. Only a careful examination of the considered set of proteins can lead to a putative statement on the respective pathway.

Thank you for your insightful input that increase in abundance of proteins associated with a pathway does not necessarily imply increase in activity of the pathway. Now we write that significant increase in abundance of proteins of a pathway only suggests importance in hypothermic adaptation or the role of RBM3 overexpression. In the specific case of RBM3 overexpression where proteins involved in ETC complex assembly (NDUF proteins), we also validated the increase in mitochondrial activity using Seahorse assay (Fig. 5A-B). We have now edited the corresponding results and discussion sections accordingly and highlighted the changes in the manuscript.

4. Lines 221-222 : CIRBP is also increased at 12h and 24h

In the latest version of our manuscript, we only focused on acute (6 hours) and chronic (48 hours) hypothermic stress. In these two cases, we observed that levels of RBM3 were increased at both acute and chronic hypothermic stress, whereas levels of CIRBP was not increased by acute hypothermic stress. Previously in figure 4A and S2I we had shown four different time-points (6, 12, 24, and 48 hours) where levels of RBM3 increased in all the time points whereas levels of CIRBP increased in 12, 24, and 48 hours. In the current version of manuscript, we have only shown levels of proteins in acute (6 hours) and chronic (48 hours) hypothermic stress. (Line 219-222)

Fig 4A

Fig S2I

5. Line 244: (fig4D, E): there is a lonely star in the figure 4 panel E (comparison between D0 and D6 RBM3 siRNA)

The star indicated an individual data point. We have updated those data points as small circle. (Revised Figure 4 panel E line 241-242)

Fig 4E

6. Lines 267-268 (fig5A-B): On panel A, red squares (maximum respiration in RBM-GFP conditions) are below the 150 pMol/min mark, whereas in panel B right, the corresponding bar indicates a mean between 150 and 200 pMol/min.

Thank you for bringing this to our attention. we have carefully re-examined the data points, leading to revisions in Figure 5A-B to ensure accuracy (line 264-270).

Fig 5A

Fig 5B

7. Line 324: it is not clear from the text, legend and mat&meth section whether the cells were transfected or infected, depending on the use of plasmids or viruses to overexpress the recombinant protein.

Thank you for bringing this to our notice. We have edited results, mat&meth section as following:

- i. All the experiments involving C2C12 cells overexpressing RBM3 was done by retro-viral transduction of C2C12 cells (Revised Line no. 265-267; Revised Materials and Methods: Line 837-838).
- ii. All the experiments of mice primary muscle cells were done by electroporating the cells with pMIG-RBM3 and pMIG-GFP plasmids (Revised Line no. 325-326; Revised Materials and Methods: Line 1026-1028).

8. Lines 335-336: same as above. What is an increase in RNA metabolism, when it can be a change in RNA synthesis, steady state, or degradation? Only a careful examination of the protein list and their functions (biological processes) can help to hypothesise the biological consequences.

As highlighted previously that increase in abundance of proteins associated with a pathway does not necessarily imply increase in activity of the pathway. Since RNA metabolism is a multi-faceted process, we have now carefully examined the protein lists and their functions. The protein-protein interactions revealed significant interactions among proteins involved in multiple aspects of RNA biology as follows-

- (i) RNA Splicing (SRRM2, SNRPB2, SF3B4, SNRPG, SNRNP200, SF3B5)
 - (ii) Cleavage and Polyadenylation (CPSF2)
 - (iii) Transcription initiation and regulation (DHX36, DHX9, HNRNPD)
 - (iv) Translation (RPS27, RPS27A, RPS27L, RPS15, RPL5, RPL8, RPL22L1)
 - (v) RNA degradation via Proteasome complex (PSMB7, PSMD6, PSMD14, PSMD8, CUL1, PSME4, PSME2)
- (Line no. 346-350)

9. Line 346 : why up- and down-regulated proteins are split into main and supplementary figures?

Based on this feedback, we have now included the down-regulated proteins in the main fig. 7. And we discussed this figure in the results and discussion sections (line 351-358).

Fig 7

(A)

(B)

General comments:

1. It would have been informative to compare lists of differentially expressed proteins in cold stress (myoblasts exposed to 32°C) and in RBM3 overexpressing myoblast. How the two lists overlap could help to better determine the role of RBM3 in cold stress response, and better characterize RBM3 direct targets.

Acknowledging this comment, we have included a figure showing overlap of the enriched pathways during chronic hypothermia and RBM3 overexpression in supplementary figure S6 E, F. Same has been included in text in line 360-363. Same has been added in supplementary figure legends (line 1147-1151).

2. The discussion section is too long with too many repetitions of results.

Acknowledging this suggestion, we have edited the discussion section in our manuscript.

3. The experiments were mainly performed in myoblasts (cold stress and/or RBM3 under-/overexpression). As myoblasts only represent a tiny percentage of the skeletal muscle tissue, could the authors justify their choice to explore the impact of cold stress in myoblasts instead of myotubes?

We agree that majority of the skeletal muscle tissue are myofibers. But this study focuses on the myoblasts because we studied the ability of muscle cells to differentiate into myotubes in response to cold stress and RBM3 under and overexpression. During differentiation myoblasts play the primary role. Therefore, we only studied these cells.